# Distinct neural bases of subcomponents of the attentional blink

**Swagata Halder[1]\*, Deepak Velgapuni Raya[1], Devarajan Sridharan[1,2]\***

[1]Centre for Neuroscience, Indian Institute of Science, Bangalore, India; [2]Computer Science and Automation, Indian Institute of Science, Bangalore, India

## eLife Assessment

This study provides an **important** contribution to our understanding of the mechanisms underlying the limited capacity to process rapid sequences of visual stimuli. It reports **convincing** evidence that the attentional blink affects neurally separable processes of visual detection and discrimination. The study will be of interest to neuroscientists and psychologists investigating perception and attention.

**Abstract** The attentional blink reflects a ubiquitous bottleneck with selecting and processing the second of two targets that occur in close temporal proximity. An extensive literature has examined the attention blink as a unitary phenomenon. As a result, which specific component of attention – perceptual sensitivity, choice bias, or both – is compromised during the attentional blink, and their respective neural bases, remains unknown. Here, we address this question with a multialternative task and novel signal detection model, which decouples sensitivity from bias effects. We find that the attentional blink impairs specifically one component of attention – sensitivity – while leaving the other component – bias – unaffected. Distinct neural markers of the attentional blink were mapped onto distinct subcomponents of the sensitivity deficits. Parieto-occipital N2p and P3 potential amplitudes characterized target detection deficits, whereas long-range high-beta band (20–30 Hz) coherence between frontoparietal electrodes signaled target discrimination deficits. We synthesized these results with representational geometry analysis. The analysis revealed that detection and discrimination deficits were encoded along separable neural dimensions, whose configural distances robustly correlated with the neural markers of each. Overall, these findings provide detailed insights into the subcomponents of the attentional blink and reveal dissociable neural bases underlying its detection and discrimination bottlenecks.

**\*For correspondence:**
swagata@iisc.ac.in (SH);
sridhar@iisc.ac.in (DS)

**Competing interest:** The authors declare that no competing interests exist.

## Introduction

Attention is a remarkable cognitive capacity that enables us to process relevant information and filter out irrelevant information to guide behavior. Yet, attention is surprisingly capacity limited (***Desimone and Duncan, 1995***). This limitation is clearly revealed when we seek to perform multiple tasks simultaneously or to tackle multiple goals in rapid succession (***Kanwisher and Potter, 1990***). A prime example of this capacity limitation is the phenomenon of the attentional blink. When multiple targets are presented sequentially – in a rapid serial visual stream – individuals are often unable to accurately detect and identify the second of two targets, particularly when it is presented in close temporal proximity (200–500ms) to the first (***Raymond et al., 1992***; *Figure 1*, left). The inability to process the second target has been hypothesized to arise from various factors, including an inherent delay in reallocating attention from the first target (T1) to the second target (T2) (***Nieuwenstein et al., 2009***; ***Vul et al., 2008***), a transient bottleneck in working memory while processing the first target (***Shapiro et al., 1994***; ***Shapiro et al., 1997***; ***Martens et al., 2002***; ***Chun and Potter, 1995***; ***Potter et al., 2005***),

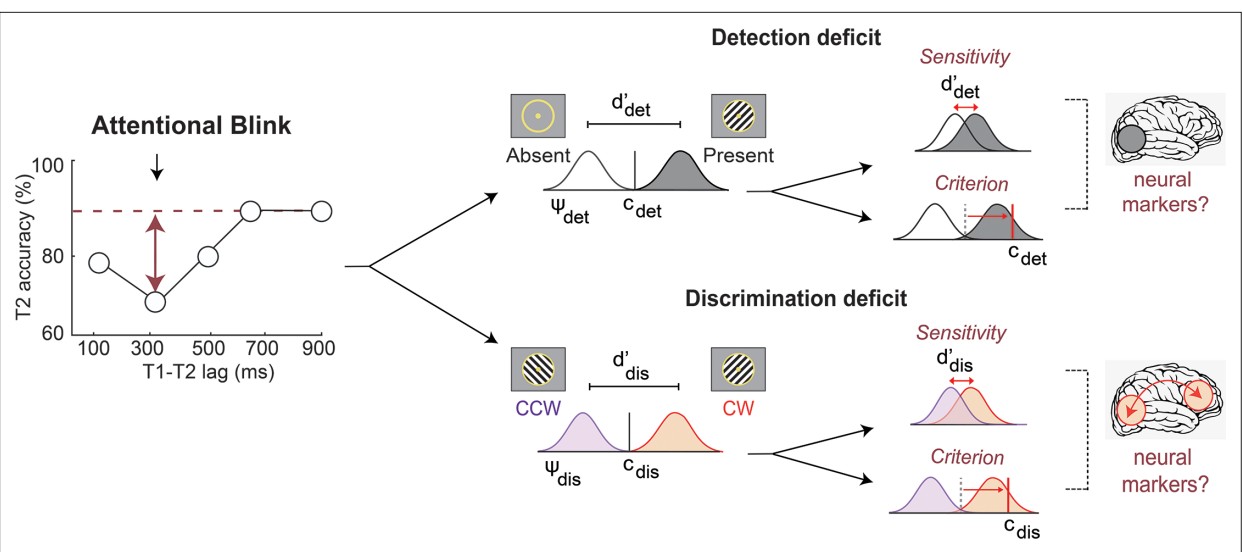

**Figure 1.** Decoupling behavioral components and neural bases of the attentional blink. (Left) Identifying the precise origin of accuracy deficits induced by the attentional blink. T2 identification accuracies at different inter-target (**T1–T2**) lags, in a conventional attentional blink task. x-axis: inter-target lag in milliseconds; y-axis: T2 identification accuracy (%). Red horizontal line: asymptotic T2 identification accuracy for long inter-target lags; red vertical arrows: accuracy deficit with T2 identification for short inter-target lags (attentional blink). (Right, top) The identification deficit could reflect impaired detection of T2's presence which, in turn, could arise either from a detection sensitivity deficit (upper row) or a detection bias (criterion) deficit (lower row). Gray and white Gaussians: decision variable distributions corresponding to signal (target present) and noise (target absent), respectively. Black vertical line: criterion for deciding between target present and absent. (Right, bottom) The identification deficit could also reflect impaired discrimination of T2's features (e.g. orientation), which, again, could arise either from a discrimination sensitivity deficit (upper row) or a discrimination bias (criterion) deficit (lower row). Purple and orange Gaussians: decision variable distributions corresponding to a counterclockwise (CW) and clockwise (CCW) gratings, respectively. Black vertical line: criterion for deciding between target features (clockwise and counterclockwise orientation). Brain schematics (rightmost column): *Distinct neural markers of each subcomponent – detection (top) or discrimination (bottom) -- of attentional blink deficits.*

or an inevitable consequence of suppressing distractors that follow the first target in the visual stream (*Di Lollo et al., 2005*; *Nieuwenstein and Potter, 2006*; *Kawahara et al., 2006*).

The effect of the attentional blink on the processing of the second target is well studied. In particular, previous studies have investigated the stage at which attentional blink affects T2's processing (early or late; *Zivony et al., 2018*; *Vogel et al., 1998*; *Nieuwenstein et al., 2005*; *Jolicoeur and Dell'Acqua, 1998*) and the neural basis of this effect, including the specific brain regions involved (*Vogel et al., 1998*; *Dell'Acqua et al., 2006*; *Luck et al., 1996*; *Sergent et al., 2005*). Several theoretical frameworks characterize a sequence of phases of the attentional blink, including target selection based on relevance, detection, feature processing, and encoding into working memory (*Chun and Potter, 1995*; *Eimer, 2014*). Overall, there is little support for attentional blink deficits at an early, sensory encoding (*Zivony et al., 2018*) stage; by contrast, the vast majority of literature suggests that T2's processing is affected at a late stage (*Martens et al., 2002*; *Potter et al., 2005*). Consistent with these behavioral results, scalp electroencephalography (EEG) studies have reported partial or complete suppression of late event-related potential (ERP) components, particularly those linked to attentional engagement (P2, N2, N2pc, or VAN; *Vogel et al., 1998*; *Dellert et al., 2022*; *Eiserbeck et al., 2024*; *Lasaponara et al., 2015*; *Vogel and Luck, 2002*), working memory (P3; *Sergent et al., 2005*; *Dell'Acqua et al., 2015*; *Chennu et al., 2009*; *Meijs et al., 2018*; *Kranczioch et al., 2003*; *Kranczioch et al., 2007*) or semantic processing (N400; *Batterink et al., 2012*); early sensory components (P1/N1) are virtually unaffected (*Sergent et al., 2005*; *Lasaponara et al., 2015*) (reviewed in detail in *Zivony and Lamy, 2022*). These 'late' effects are hypothesized to be mediated by diverse functional regions within a frontoparietal network, both based on correlational and causal evidence (*Marois et al., 2000*; *Cooper et al., 2004*; *Kihara et al., 2007*). Activity in the parietal, lateral prefrontal, and anterior cingulate cortex, as measured with functional MRI (fMRI), was shown to decrease when T2 was not detected (missed), even though stimulus-evoked activity in early visual areas (e.g. V1) was relatively intact (*Williams et al., 2008*). Long-range synchronization of EEG oscillations across fronto-parietal regions in the beta-band (13–18 Hz) is impaired during the attentional blink

(*Gross et al., 2004*). In addition, converging evidence from transcranial magnetic stimulation (TMS) studies suggests a potential role for the posterior parietal cortex in the attentional blink (*Cooper et al., 2004*; *Kihara et al., 2007*): T2 detection and identification performance were enhanced after the application of transcranial magnetic stimulation (TMS) over the bilateral parietal cortex.

Despite this extensive literature, many previous studies have studied the attentional blink as a unitary phenomenon. While some theoretical models (*Chun and Potter, 1995*; *Eimer, 2014*; *Zivony and Lamy, 2022*) and experimental studies (*Eiserbeck et al., 2024*; *Harris et al., 2013*) have explored distinct mechanisms underlying the attentional blink, several fundamental questions about its distinct component mechanisms remain unanswered. One key question concerns the precise nature of the 'late' attentional blink bottleneck. The attentional blink effect is typically quantified as a decrease in the proportion of correct T2 responses at short, relative to long, inter-target lags (*Raymond et al., 1992*). Yet, such an impairment can arise from one of at least two processes – a reduction in the fidelity of T2's perceptual representation, a deficit in decision-making based on T2's representation, or both. Signal detection theory (*Green and Swets, 1966*) is a widely applied psychophysical framework whose parameters – sensitivity (d') and bias (or criterion, c) – quantify these perceptual and decisional components, respectively.

Distinguishing between sensitivity and criterion effects is crucial because a change in either of these parameters can produce a change in the proportion of correct responses (*Luo and Maunsell, 2015*; *Sridharan et al., 2017*). A lower proportion of correct T2 detections may reflect not only a lower detection d' at short lags but also a sub-optimal choice criterion corresponding, for instance, to a conservative detection bias (*Asplund et al., 2014*; *Figure 1*, right, top). Importantly, such criterion effects need not be uniform across inter-target lags: the lag on each trial could be inferred based on various factors, such as trial length, allowing participants to adopt different choice criteria for the different lags prior to making a response (but see *Chun and Potter, 1995*). Yet, whether the attentional blink induces primarily d' deficits, primarily bias deficits, or a mixture of both is largely unknown, and the limited evidence available is controversial (*Lasaponara et al., 2015*; *Caetta and Gorea, 2010*). This is perhaps because previous studies typically employed simple target detection or identification tasks, in conjunction with conventional, one-dimensional signal detection models (*Lasaponara et al., 2015*; *Caetta and Gorea, 2010*). Recent literature suggests that such tasks and models do not suffice to reliably distinguish between sensitivity and bias components of attention (*DeCarlo, 2012*; *Sridharan et al., 2014*; *Hautus et al., 2021*). A complete understanding of the attentional blink requires dissociating sensitivity from bias deficits, with appropriate signal detection models, and identifying their respective neural correlates.

A second, and arguably more salient, question concerns the precise nature of attentional blink deficits. Is the attentional blink a deficit with detecting T2 (*Figure 1*, right, top) or discriminating T2's features, or both (*Figure 1*, right, bottom)? It is evident, from first principles, that distinct neural computations and regions must mediate target detection versus discrimination. For example, detecting an oriented grating in noise can be achieved with a simple, local visual cortex computation: first, by aggregating responses of orientation-tuned visual neurons and then, by testing if this activity exceeds a pre-set threshold (*Parker and Newsome, 1998*). By contrast, discriminating the orientation of a target grating involves an arguably more nuanced computation; one that requires comparing the relative activities of neurons tuned to different orientations relative to some predefined axis (e.g. vertical meridian; *Parker and Newsome, 1998*). This latter computation would also involve higher cortical areas for maintaining the feature discrimination rule in working memory, such as the prefrontal cortex (*Funahashi, 2017*; *Zhang et al., 2023*; *Dupont et al., 1998*). Moreover, each of these computations could be affected by attention differently (*Maunsell, 2015*).

This second question is particularly important because conventional attentional blink task designs often conflate these computations. In a typical attentional blink task, participants must make identification judgments on T2 (*Raymond et al., 1992*; *Shapiro et al., 1994*). For example, participants may be asked to identify T2 based on its shape (e.g. a specific letter among numbers) or its category (e.g. a face among non-faces). Yet, an identification deficit during the attentional blink could arise from at least one of two sources (*Raymond et al., 1992*; *Nieuwenstein et al., 2009*; *Chun and Potter, 1995*; *Vogel and Luck, 2002*; *Gross et al., 2004*). First, the blink could produce a detection deficit, when the participant fails to reliably detect the occurrence of T2 (*Figure 1*, Top Right). Alternatively, the blink could produce a discrimination deficit, when the participant detects T2, but is unable to reliably

discriminate its features (*Figure 1*, Bottom Right). A combination of both of these deficits is also possible. To determine the precise neural underpinnings of the attentional blink deficits, these two deficits must be dissociated, and their respective neural correlates identified.

To tease apart these distinct subcomponents of the attentional blink, we developed a multialternative task that involved concurrent detection of T2 and discrimination of its features (*Figure 2A*). To analyze multialternative behavioral responses, we developed a novel two-dimensional signal detection model that decoupled and separately quantified sensitivity and bias deficits during the blink. With concurrent EEG recordings, we analyzed key neural markers – a local neural marker (event-related potentials) and a global neural marker (oscillatory frontoparietal synchronization) – to test if these would map onto distinct subcomponents of the attention blink (detection and discrimination deficits, respectively). We synthesized these results with representational geometry analysis to understand the neural representations underlying subcomponents of attentional blink-induced deficits. Our results reveal a double dissociation between the neural and behavioral bases of detection and discrimination bottlenecks underlying the attentional blink.

## Results

### The attentional blink produces both detection and discrimination deficits

Participants (n=24) performed an attentional blink task involving target detection and orientation discrimination (*Figure 2A*, see also Methods). In a stream of serially flashed stimuli (100ms each), the first target (T1) was an oriented grating (radius: 2 degrees of visual angle or dva) of low spatial frequency (0.6 cycles per degree or cpd), whereas the second target (T2) was either an oriented grating of comparatively higher spatial frequency (1.8 cpd, 67% of trials) or a blank (33% of trials). T2 stimuli of high contrast (100%) and low contrast (titrated to individual threshold, see Materials and methods) were interleaved with equal (50%) probability across trials. Targets were interspersed with plaid distractors, and inter-target onset intervals (T1-T2 'lag'-s) were drawn from a geometric distribution (100 ms to 900 ms; see Materials and methods).

Unlike conventional attentional blink tasks (*Raymond et al., 1992*; *Shapiro et al., 1994*; *Sergent et al., 2005*), a key element of novelty in our task design was the following: for T1 participants provided a two-alternative orientation judgement, whereas for T2 they provided a three-alternative, concurrent detection and identification judgement (*Figure 2A*). Specifically, at the end of each trial, participants first indicated whether they perceived T1's orientation to be closer to the cardinal (0°, ±90°) or the diagonal axes (±45°) with two distinct button-press responses (two-alternative forced choice or 2-AFC; *Figure 2A*, penultimate panel from the right). Then, participants reported whether they had detected T2, and if so, whether its orientation was clockwise or counterclockwise of vertical, using 3 distinct button-press responses (three-alternative forced choice or 3-AFC; *Figure 2A*, rightmost). The 3-AFC decision enabled us to decouple the effect of the attentional blink on detection and discrimination performance.

First, we estimated detection and discrimination accuracies for T2 judgements across different T1-T2 lags; these analyses were performed only on trials in which participants provided accurate T1 responses (denoted as 'T2|T1' in the figures). Detection accuracies were calculated based on the proportion of trials in which T2 was correctly detected (Materials and methods). Briefly, we computed the average proportion of hits, misidentifications, and correct rejections; misidentifications were included because, although incorrectly identified, the target was nevertheless correctly detected. In contrast, discrimination accuracies were derived from T2 present trials, based on the proportion of correct identifications alone (Materials and methods). T2 detection accuracies were markedly lower at the short (100ms and 300ms), as compared to the long, inter-target lags (700 and 900ms; *Figure 2B*); this induced a significant detection deficit, measured as the difference in detection accuracies between the short and the long lags (detection deficit = −8.9 ± 1.6%, mean ± s.e.m., z=−4.14, p<0.001, Wilcoxon signed-rank test, BF >$10^2$; data pooled across T2 contrasts). Similarly, T2 discrimination accuracies were also significantly lower at the short lags compared to the long lags (discrimination deficit = −13.5 ± 1.8%, z=−4.25, p<0.001, BF >$10^2$; *Figure 2C*). In other words, the attentional blink induced both a significant detection and a discrimination deficit for the processing of the second target.

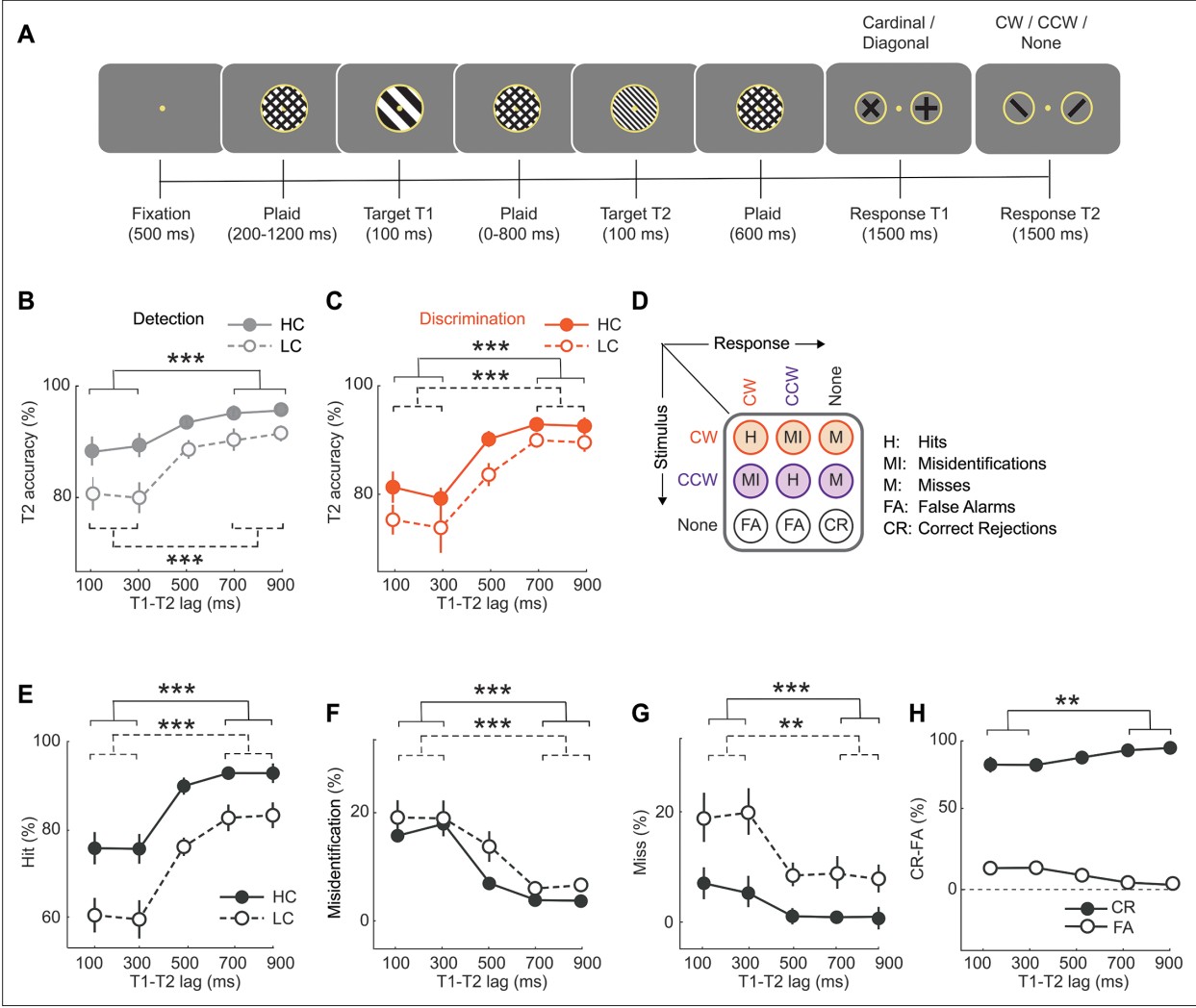

**Figure 2.** Novel task design to distinguish subcomponents of the attentional blink. (**A**) Schematic of the attentional blink task. Stimuli were presented in a rapid serial visual presentation (RSVP) paradigm at a 10 Hz rate (70ms onset, 30ms offset). Following fixation, plaid gratings appeared for a variable interval (200–1200ms, geometrically distributed), followed by the first target (**T1**): a low spatial frequency grating (100ms). After this, a series of plaid gratings appeared for variable intervals (100, 300, 500, 700, and 900ms; geometric distribution) followed by the appearance of the second target (**T2**): a high spatial frequency grating (100ms). Following T2, plaid gratings were presented for a fixed interval (600ms). Finally, in the response epoch, participants reported T1's orientation as being closer to the cardinal or diagonal axes (two-alternative), and then reported T2's orientation as being clockwise or counterclockwise of vertical, or absent (three-alternative). All plaids were encircled by a circular placeholder. The fixation dot and the placeholder were present on the screen throughout the trial. (**B**) Psychometric function of accuracy (% correct) for T2 detection with increasing inter-target (**T1–T2**) lags, for trials in which T1 was reported correctly (n=24 participants). Filled circles and solid lines: average accuracy for high contrast T2 gratings; open circles and dashed lines: average accuracy for low contrast T2 gratings. Error bars: s.e.m. Asterisks: significance levels for comparing accuracies between short (100 and 300ms) and long (700 and 900ms) lag trials; Statistical method: Wilcoxon signed-rank test; solid and dashed brackets: comparisons for high and low contrast gratings respectively. *p<0.05, **p<0.01, ***p<0.001 and n.s.: not significant. (**C**) Same as in panel B, but showing the psychometric function of accuracy for T2 discrimination with increasing inter-target (**T1–T2**) lags (n=24). Other conventions are the same as in panel B, except that markers and lines are depicted in orange color. (**D**) Stimulus-response contingency table for the 3-alternative T2 decision. Rows represent the three possible T2 stimulus events: clockwise orientation (CW, orange), counterclockwise orientation (CCW, purple) or absent (none, gray). Columns represent three possible choices: clockwise (CW), counterclockwise (CCW) or absent (none). The table depicts the nine stimulus-response contingencies: two each of hit rates (H), misidentification rates (MI), miss rates (M), false alarm rates (FA) – one for each orientation (CW/CCW) – and one correct rejection rate (CR). (**E**) Same as in panel B, but showing psychometric function of average hit rates. (**F**) Same as in panel B, but showing psychometric function of average misidentification rates. (**G**) Same as in panel B, but showing psychometric function of average miss rates. (**E–G**) Other conventions are the same as in panel B except that markers and lines are denoted in black color. (**H**) Same as in panel B, but showing psychometric function correct rejection (filled circles) and false alarm (open circles) rates on T2 absent trials.

Next, we tested if the magnitude of either the detection or the discrimination deficit would vary depending on T2 stimulus contrast. For this, we performed a two-way ANOVA with either the detection or the discrimination accuracy as the response variable and lags (short or long) and contrast (high/HC or low/LC) as independent factors. While we found a main effect of both lag (detection: $F_{(1,23)}=29.8$, $p<0.001$, BF $>10^3$, discrimination: $F_{(1,23)}=54.1$, $p<0.001$, BF $>10^3$) and contrast (detection: $F_{(1,23)}=21.02$, $p<0.001$, BF $>10^2$, discrimination: $F_{(1,23)} = 13.75$, $p=0.001$, BF = 1.2), we found no significant interaction effect between lag and contrast (detection: $F_{(1,23)}=1.92$, $p=0.113$, BF = 0.49, discrimination: $F_{(1,23)} = 0.93$, $p=0.450$, BF = 0.40). In other words, attentional blink induced both a detection and discrimination deficit regardless of stimulus contrast.

Finally, in addition to analyzing deficits with overall accuracies, we also analyzed blink-induced effects on individual behavioral responses in the 3x3 stimulus-response contingency table for the 3-AFC task (*Figure 2D*). These responses fall under five categories – hits, correct rejections, misidentifications, false alarms, and misses (Materials and methods). Again, a two-way ANOVA revealed a main effect of inter-target lag for all five response types ($p<0.05$; BF $>10$, *Supplementary file 1A*): essentially all correct response proportions (hits, correct rejections) decreased, and incorrect response proportions (false alarms, misses, and misidentifications) increased, at short relative to long inter-target lags. A main effect of T2 contrast was observed for hit and miss proportions ($p<0.001$) but not for misidentification proportions ($p=0.090$; *Supplementary file 1A*); note that, by definition, false alarms and correct rejection proportions are independent of T2 contrast.

Overall, the attentional blink affected both components of identification – detection and discrimination – in terms of overall accuracy and individual psychometric measures. Whereas detection and discrimination accuracies varied with T2 stimuli's contrast, the strengths of detection and discrimination deficits induced by the blink did not depend on contrast.

## Attentional blink selectively impairs the sensitivity, but not the bias subcomponent

While the attentional blink induced deficits with both T2 detection and discrimination, such deficits could arise from either sensitivity or bias/criterion mechanisms. These two possibilities are illustrated in *Figure 1*, using a one-dimensional, signal detection theory (1-D SDT) model. In the first case (*Figure 1*, right, top, upper row), a deficit in T2 detection occurs because of a deterioration in T2's signal evidence strength, which manifests behaviorally as lower perceptual sensitivity. In the second case (*Figure 1*, right, top, lower row), the detection deficit occurs because of a more conservative evidence threshold for reporting the presence of T2, which manifests, behaviorally, as a higher criterion, or a lower bias. Either (or both) of these effects could induce an accuracy deficit (*Figure 1*, left). Similarly, discrimination deficits can occur either because of impaired sensitivity (*Figure 1*, right, bottom, upper row) or a sub-optimal bias (*Figure 1*, right, bottom, lower row). We sought to distinguish the precise source – sensitivity versus bias – of deficits induced by the attentional blink.

To address this question, we developed a novel, two-dimensional signal detection model to analyze participants' T2 judgments in the 3-AFC task (see Discussion). Note that such multialternative responses cannot be correctly analyzed with a combination of one-dimensional SDT models; the reasons are discussed in detail in previous studies (*Sridharan et al., 2017*; *Sridharan et al., 2014*; see also Discussion). Briefly, the three-alternative decision is modeled in a two-dimensional decision space: evidence for the presence (or absence) of T2 represented along the abscissa and evidence for clockwise or counterclockwise orientations for T2 represented along the ordinate (*Figure 3A*); we term these the 'detection' (y-axis) and 'discrimination' (x-axis) axes, respectively. On each trial, a bivariate decision variable ($\psi$) encodes T2's features, with $\psi$'s component along the detection and discrimination axes representing evidence for T2's presence ($\phi_{det}$) and T2's orientation ($\phi_{dis}$), respectively. The distribution of $\psi$, across trials, is modeled as a bivariate, isotropic Gaussian whose mean varies with T2's configuration across each of five conditions (*Figure 3A–B*; absent, LC/CW, LC/CCW, HC/CW, HC/CCW); thus, the model incorporates one noise distribution (T2 absent), centered at the origin, and four signal distributions. The means of the signal distributions along the detection and discrimination axes represent the participant's sensitivity for detecting T2 ($d'_{det}$) and discriminating its orientation ($d'_{dis}$), respectively. An inverted-Y shaped decision surface divides the decision space into 3 decision zones, one corresponding to each type of response: T2 clockwise (upper right), T2 counterclockwise (upper left) or T2 absent (lower). The decision surface is parameterized by a detection threshold ($t_{det}$)

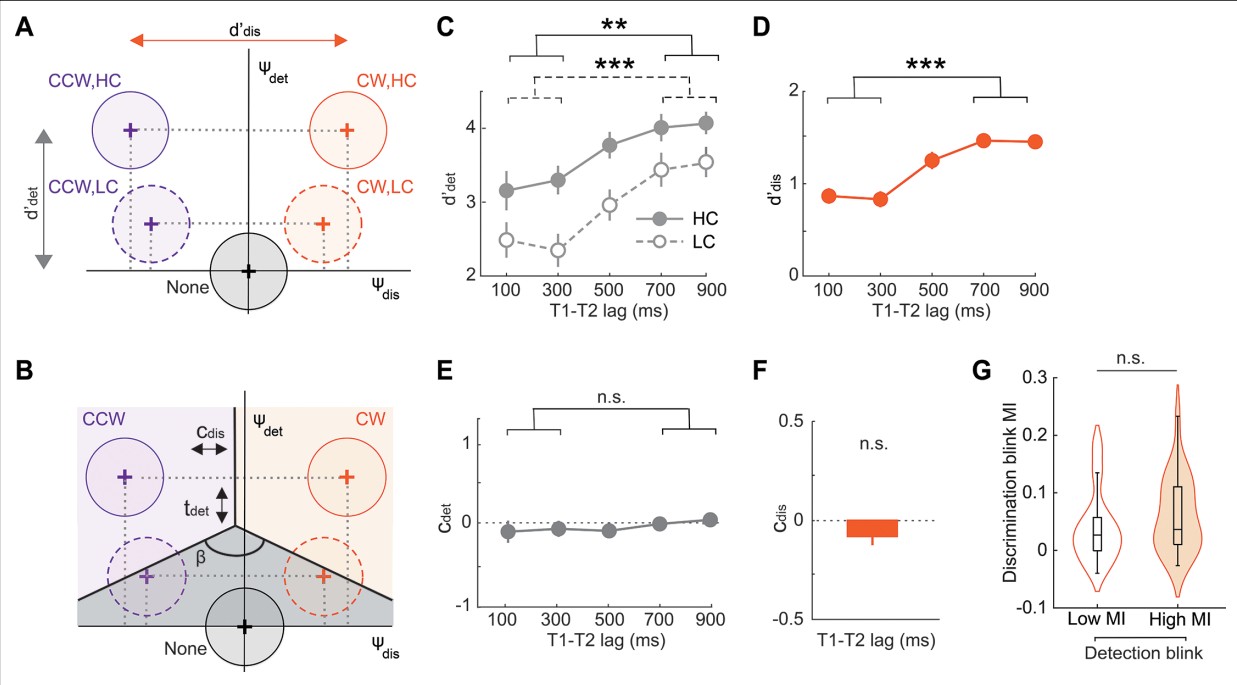

**Figure 3.** A novel psychophysical model decouples attentional blink subcomponents. (**A**) Signal detection model for estimating T2 sensitivity and bias. The decision variable is a bivariate Gaussian ($\Psi$) whose components, $\Psi_{det}$ (ordinate) and $\Psi_{dis}$ (abscissa), encode sensory evidence for stimulus presence and orientation, respectively. Black circle: $\Psi$ distribution for T2 absent trials (noise distribution) is centered on the origin. Orange and purple circles: $\Psi$ distributions for clockwise and counterclockwise T2 orientations, respectively (signal distributions). Solid and dashed outlines: High and low contrast T2, respectively. Vertical gray and horizontal orange lines (double-headed arrows): detection sensitivity ($d'_{det}$) and discrimination sensitivity ($d'_{dis}$) for high contrast T2, respectively. Dashed gray lines: Signal mean projections onto the detection and discrimination axes. (**B**) Decision surface with linear decision boundaries (thick black lines) demarcates 3 decision zones, one for each potential 3AFC choice: Clockwise T2 (CW, orange shading), counterclockwise T2 (CCW, purple shading), no T2 (None, gray shading). The decision surface is parameterized by: (i) a discrimination criterion ($c_{dis}$), governing the horizontal position of the decision surface (horizontal double arrowhead), (ii) a detection threshold ($t_{det}$), governing the vertical position of the decision surface (vertical double arrowhead), and (iii) the angle between the two oblique decision boundaries ($\beta$). Other conventions are the same as in panel A. (**C**) Psychophysical function of detection sensitivity ($d'_{det}$) with increasing inter-target (**T1–T2**) lags. Other conventions are the same as in *Figure 2B*. (**D**) Same as in panel C but showing the psychophysical function of discrimination sensitivity ($d'_{dis}$). Model selection yielded identical psychophysical functions for high and low contrast T2 (Materials and methods). Other conventions are the same as in *Figure 2B*. (**E**) Same as in panel C but showing the psychophysical function of detection criterion ($c_{det}$). Dashed horizontal line: $c_{det} = 0$. (**F**) Discrimination criterion ($c_{dis}$). Model selection constrained $c_{dis}$ to be equal across lags (Materials and methods). Other conventions are the same as in panels D-E. (**G**) Modulation index (MI) of the discrimination blink for low (open plot) and high (filled plot) detection blink MI blocks. Box plots, center line: median; box limits: upper and lower quartiles; whiskers: 1.5 x the interquartile range. Violin plots: kernel density estimates. (**C–G**) Statistical method: Wilcoxon signed-rank test; Asterisks: *p<0.05, **p<0.01, ***p<0.001; and n.s.: not significant.

The online version of this article includes the following figure supplement(s) for figure 3:

**Figure supplement 1.** Model comparison analysis.

**Figure supplement 2.** Correlation between detection and discrimination blink.

and a discrimination criterion ($c_{dis}$); these parameters determine the placement of the decision surface along the detection and discrimination axes, respectively. In addition, the angle between the lower arms of the decision surface ($\beta$, *Figure 3B*) is estimated as a free parameter of the model. The model was fit to the 3x3 stimulus-response contingency table derived from participants' 3-AFC responses (Materials and methods) with maximum likelihood estimation.

We fit five variants of the model with different constraints on the parameters based on distinct sets of plausible assumptions: these included constraints on the parameters across lags, as well as across low and high contrast T2 stimuli (Materials and methods; *Figure 3—figure supplement 1A*). Model comparison analysis – based both on the Akaike Information Criterion (AIC) *Vrieze, 2012* and the Bayesian information criterion (BIC) *Vrieze, 2012* – identified a model in which discrimination sensitivity ($d'_{dis}$) was equal for high and low contrast T2 stimuli for each inter-target lag, and discrimination criterion ($c_{dis}$) and angle ($\beta$) were equal across lags (Model III, *Figure 3—figure supplement 1A*, C-D).

Goodness-of-fit p-values (Materials and methods) were generally high for this model (median >0.95; *Figure 3—figure supplement 1B*), indicating satisfactory model fits to data.

The attentional blink significantly impaired both detection and discrimination sensitivity. T2 detection d' was significantly lower at short, compared to long, inter-target lags (d'$_{det}$ deficit = –0.94 ± 0.17, z=−3.80, p<0.001, signrank test; BF >10$^3$, data pooled across T2 contrasts; *Figure 3C*). Similarly, T2 discrimination d' was also significantly lower at the short lags compared to the long lags (d'$_{dis}$ deficit = –0.61 ± 0.06, z=−4.25, p<0.001; BF >10$^3$; *Figure 3D*). A two-way ANOVA with inter-target lag and T2 contrast as independent factors revealed a main effect of lag on both d'$_{det}$ (F(1,23)=30.3, p<0.001, BF >10$^3$) and d'$_{dis}$ (F(1,23)=100.3, p<0.001, BF >10$^3$). Yet, we found no significant interaction effect between lag and contrast for d'$_{det}$ (F(1,23)=2.3, p=0.141, BF = 0.44). A similar interaction analysis could not be performed for d'$_{dis}$ as this parameter was constrained to be equal across contrasts, based on model selection analysis.

By contrast, the attentional blink produced no systematic effect on either detection or discrimination criteria. Even though the detection threshold (t$_{det}$) was higher at short, compared to long, inter-target lags (t$_{det}$ deficit = –0.59 ± 0.17, z=−3.11, p<0.001, BF = 14), the detection criterion (c$_{det}$) – the conventional SDT measure of bias (Materials and methods) – was not statistically significantly different across lags (c$_{det}$ deficit = –0.16 ± 0.16, z=−0.28, p=0.775, BF = 0.34; *Figure 3E*). Moreover, model selection analysis had already identified a model in which the discrimination criterion (c$_{dis}$) was constrained to be invariant across lags (*Figure 3F*), obviating analyses of lag effects on this parameter. Nonetheless, estimating this criterion – even with an unconstrained model – revealed substantial evidence against a statistically significant blink effect (c$_{dis}$ deficit = 0.02 ± 0.04, z=−0.485, p=0.627, BF = 0.23).

Finally, we tested whether detection and discrimination deficits were likely to be mediated by common or dissociable processes. For this, first, we correlated the magnitude of the blink-induced detection and discrimination sensitivity deficits – the d' modulation index (MI) – across participants (Materials and methods). Detection and discrimination d' deficits were not statistically significantly correlated (*r*=0.39, p=0.059); Bayes factor analysis revealed no clear evidence for or against a correlation between these subcomponent deficits (BF = 1.18; *Figure 3—figure supplement 2*, left). Similar results were obtained upon correlating accuracy deficits also (*Figure 3—figure supplement 2*, right). In general, detection d' deficits varied widely even among individuals exhibiting a narrow range of discrimination deficits, and vice versa (*Figure 3—figure supplement 2A–B*). Second, we tested whether the strength of detection and discrimination accuracy deficits would co-vary within each participant. For this, we divided task blocks into those with the highest and lowest detection accuracy MIs and compared the magnitude of the discrimination accuracy MI across these blocks (Materials and methods). Discrimination accuracy deficits were not statistically significantly different between high and low detection accuracy deficit blocks (z=1.97, p=0.067), and the Bayes factor revealed no strong evidence for or against such a difference (BF = 1.42; *Figure 3G*).

In summary, we developed a novel, two-dimensional signal detection model to simultaneously quantify blink-induced effects on T2's detection and discrimination. The attentional blink affected both detection and discrimination sensitivity, across T2 stimuli of high and low contrasts. Yet, no measurable effect occurred on detection or discrimination criteria. In other words, performance deficits induced by the attentional blink could be attributed entirely to sensitivity, rather than criterion, effects. Moreover, detection and discrimination d' deficits were not significantly correlated both within and across participants, with no clear evidence for or against a correlation, based on the Bayes factor. Next, we investigated electrophysiological correlates of these behavioral deficits.

## Dissociable electrophysiological markers of detection versus discrimination deficits

We sought to identify electrophysiological correlates of detection and discrimination deficits during the attentional blink. As a first step, we examined the effect of established signatures of the attentional blink, including event-related potential (ERP) magnitude (*Vogel et al., 1998*; *Sergent et al., 2005*) and long-range synchronization (*Gross et al., 2004*). EEG data were acquired from a subset (n=18/24) of participants, while they were tested on the behavioral paradigm (Materials and methods); high and low contrast T2 trials were pooled to estimate reliable ERPs (Materials and methods). To prevent confounding neural correlates of detection and discrimination d' deficits, we quantified the

partial correlation ($r_p$) between the amplitudes of each EEG metric and either detection d' ($d'_{det}$) or discrimination d' ($d'_{dis}$) across lags, while controlling for the value of the other parameter ($d'_{dis}$ or $d'_{det}$, respectively; Materials and methods).

First, we quantified the change in peak amplitude for different ERP components conventionally associated with the attentional blink: parietal P1 (*Sergent et al., 2005*; *Lasaponara et al., 2015*), frontocentral P2 (*Vogel et al., 1998*; *Akyürek et al., 2010*), occipital N2p (*Vogel et al., 1998*; *Sergent et al., 2005*; *Lasaponara et al., 2015*; *Kranczioch et al., 2007*) and parietal P3 (*Vogel et al., 1998*; *Dell'Acqua et al., 2006*; *Jolicœur et al., 2006*; *Jolicoeur et al., 2006*; *Figure 4A–B*, left). Among these ERP components, the N2p component and the P2 component were both significantly suppressed during the blink (Δamplitude, short-lag – long-lag: N2 p=−0.47 ± 0.12 µV, z=−3.20, p=0.003, BF = 40.0, P2=−0.19 ± 0.07 µV, z=−2.54, p=0.021, BF = 4.83, signed rank test; *Figure 4A*, right). Similarly, the parietal P3 also showed significant blink-induced suppression (P3=−0.45 ± 0.09µV, z=−3.59, p<0.001, BF >$10^2$; *Figure 4B*, right). By contrast, the P1, an early sensory component, showed no statistically significant blink-induced modulation (P1=0.25 ± 0.16µV, z=1.19, p=0.231, BF = 0.651; *Figure 4—figure supplement 1*). Results from various studies support each of these findings (*Sergent et al., 2005*). In addition, we did not observe a statistically significant N1 component in the long lag trials (p=0.207, one-tailed signed rank test for amplitude less than zero); the Bayes factor (BF = 1.35) revealed no clear evidence for an N1 component.

Next, we tested whether distinct ERP components correlated specifically with detection versus the discrimination deficits induced by the attentional blink. Detection d' correlated both with the N2p and P3 amplitudes (*Figure 4C and E*; partial correlation, N2p: $r_p$ = 0.34 p<0.001, P3: $r_p$ = 0.30, p<0.001, permutation test; *Figure 4D and F*, top). However, discrimination d' correlated with neither of these components (N2p: $r_p$ = −0.01, p=0.970; P3: $r_p$ = 0.14, p=0.748; *Figure 4D–F*, bottom). Yet, despite its amplitude being significantly modulated by the blink, the frontocentral P2 component did not correlate with either detection d' ($r_p$ = 0.05, p=0.999) or discrimination d' ($r_p$ = 0.23, p=0.120). Similarly, the P1 component did not correlate significantly with either detection or discrimination d' (full set of results in *Supplementary file 1B*). In other words, two key late ERP components (N2p and P3) correlated with detection d' deficits, but not with discrimination d' deficits, each after controlling for the variance explained by the other variable.

Finally, we tested whether long-range synchronization across the brain correlated specifically with blink-induced detection or discrimination deficits. Following previous reports (*Kranczioch et al., 2003*; *Gross et al., 2004*), we investigated synchronization in the beta band (13–30 Hz) between frontal and parietal electrodes using spectral coherence (Materials and methods). We observed a strong and sustained decrease in fronto-parietal coherence during the attentional blink, particularly in the high-beta (20–30 Hz) band (*Figure 5A–B*, ΔPLV=PLV$_{short-lag}$–PLV$_{long-lag}$). Hemisphere-wise analysis revealed a marked and significant reduction in fronto-parietal high-beta coherence over the left hemisphere, in a cluster from 0 to 300ms post T2 onset (permutation test, p=0.038, corrected; cluster-forming threshold p<0.05; *Figure 5C*); by contrast, no discernible effects occurred over the right hemispheric fronto-parietal electrodes (p>0.05; *Figure 5D*). Furthermore, we tested whether high-beta coherence varied with detection d' or discrimination d', lag-wise, with partial correlations (Materials and methods). Discrimination d', but not detection d', was significantly correlated with left fronto-parietal coherence in the high-beta band ($d'_{dis}$: $r_p$ = 0.22, p=0.018; $d'_{det}$: $r_p$ = −0.05, p=0.316; *Figure 5E–F*).

We also examined correlations between detection d' or discrimination d' and low-beta (13–19 Hz) coherence, as well as with beta coherence over right hemispheric fronto-parietal electrodes but found no significant results; the full set of correlations is shown in *Supplementary file 1C*.

To summarize, we observed a remarkable double dissociation between electrophysiological markers of detection and discrimination deficits induced by the attentional blink: whereas N2p and P3 suppression signaled blink-induced detection d' deficits, reduction in high-beta phase synchronization between left fronto-parietal electrodes signaled discrimination d' deficits.

## Detection and discrimination deficits map onto distinct neural dimensions

Recent studies suggest that an analysis of neural population geometry may provide mechanistic insights into diverse cognitive phenomena (*Sheahan et al., 2021*; *Chung and Abbott, 2021*). Inspired by these findings, we tested whether behavioral detection and discrimination deficits could

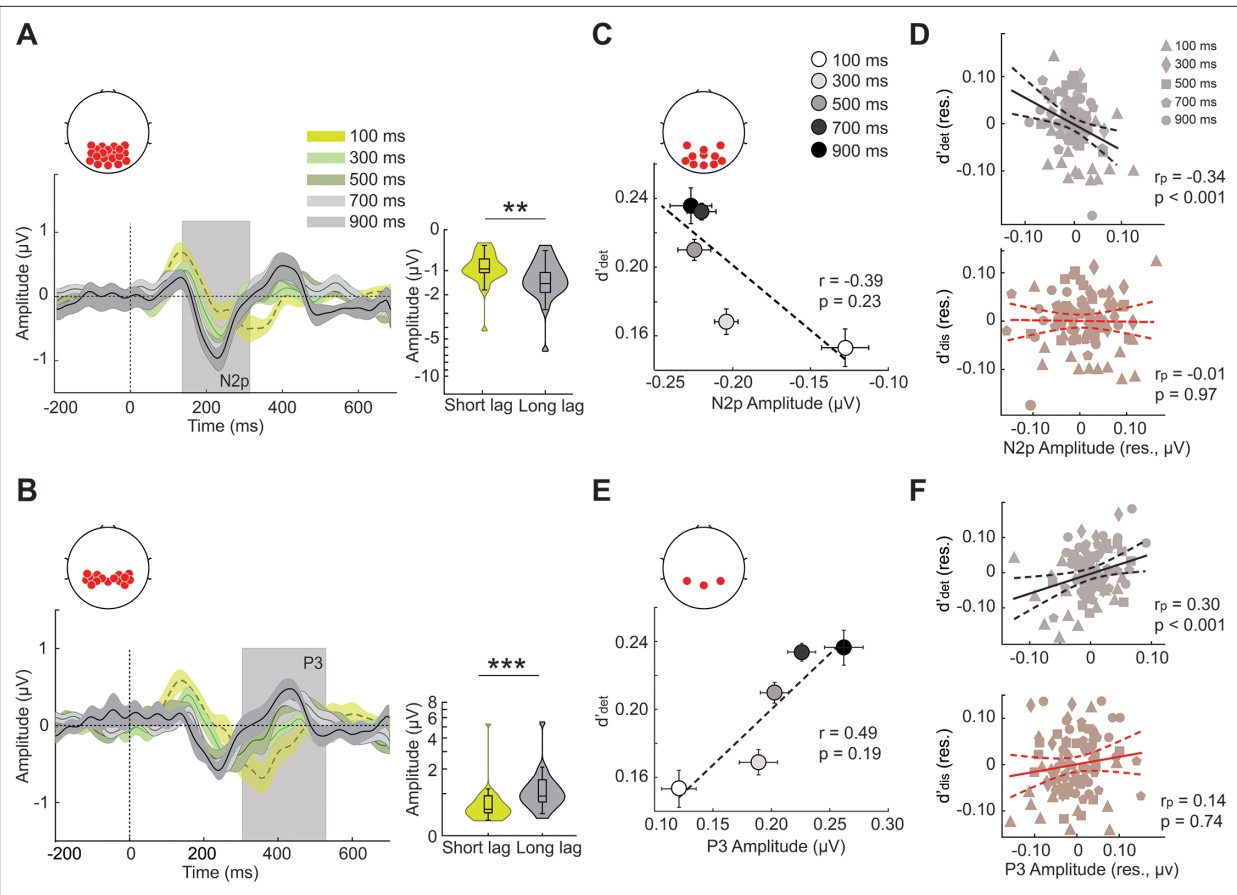

**Figure 4.** N2p and P3 event-related potentials signal detection sensitivity deficits. (**A**) (Left) The event-related potential (ERP, n=18 participants) showing the N2p component in occipitoparietal electrodes (inset), locked to T2 onset (dashed vertical line). Bright green, dull green, dark green, light gray, and black traces: Average ERPs for the five inter-target lags – 100, 300, 500, 700, and 900ms – respectively. Shading: s.e.m. Gray vertical shading: Time epoch for N2p amplitude quantification. ERPs were computed by subtracting the average ERPs on correct rejection trials (Materials and methods). (Right) Violin plots showing the distribution of the peak N2p amplitudes across participants separately for the short (green: 100+300ms) and the long (gray: 700+900ms) lag trials. Asterisks: *p<0.05, **p<0.01, and ***p<0.001. Other conventions are the same as in *Figure 3G*. (**B**) (Left) Same as in panel A (left) but showing the P3 ERP component in the parietal electrodes (inset). (Right) Same as in panel A (left) but showing the quantification of the P3 ERP. Other conventions are the same as in panel A. (**C**) Variation of detection sensitivity (d'det; y-axis) with N2p peak amplitude (x-axis) across inter-target lags (circles). r and p denote the robust correlation coefficient and permutation test p-value, respectively (Materials and methods). Dashed line: linear fit. Error bars: s.e.m. (**D**) (Top) Partial correlation between N2p peak amplitude (x-axis, amplitude residual) and detection sensitivity (y-axis, d'det residual) while controlling for the discrimination sensitivity (d'dis). Each of the five shapes – filled triangle, diamond, square, pentagon, circle – represents one inter-target lag (legend). rp and p denote the partial correlation coefficient and permutation test p-value, respectively (Materials and methods). Solid line: Linear fit; dashed curves: 95% confidence intervals. (Bottom) Same as in the top panel but showing partial correlation between N2p peak amplitude (x-axis, amplitude residual) and discrimination sensitivity (y-axis, d'dis residual) while controlling for detection sensitivity (d'det). (**E**) Same as in panel C but showing variation of discrimination sensitivity (d'dis, y-axis) with P3 peak amplitude (x-axis) across inter-target lags. (**F**) Same as in panel D but showing the partial correlation of P3 peak amplitude with d'det while controlling for d'dis (top), or, conversely, with d'dis while controlling for d'det (bottom). (**E–F**) Other conventions are the same as in panels C-D, respectively.

The online version of this article includes the following figure supplement(s) for figure 4:

**Figure supplement 1.** P1 ERP component in parietal electrodes.

be mapped to distinct neural dimensions. For this, we quantified the pairwise Euclidean distance between the centroids of multivariate activity patterns produced by each class of T2 stimuli (clockwise, counterclockwise, or no grating) in the posterior electrodes, separately for each inter-target lag (Materials and methods). With these distances, we identified (i) a 'detection' dimension ($\eta_{det}$), a neural dimension that encoded the presence versus absence of T2 and (ii) a 'discrimination' dimension, ($\eta_{dis}$) encoded T2's orientation (clockwise versus counterclockwise; Materials and methods; *Figure 6A–B*, left). Neural inter-class distances ($\| \eta \|$) along both the detection and discrimination dimensions

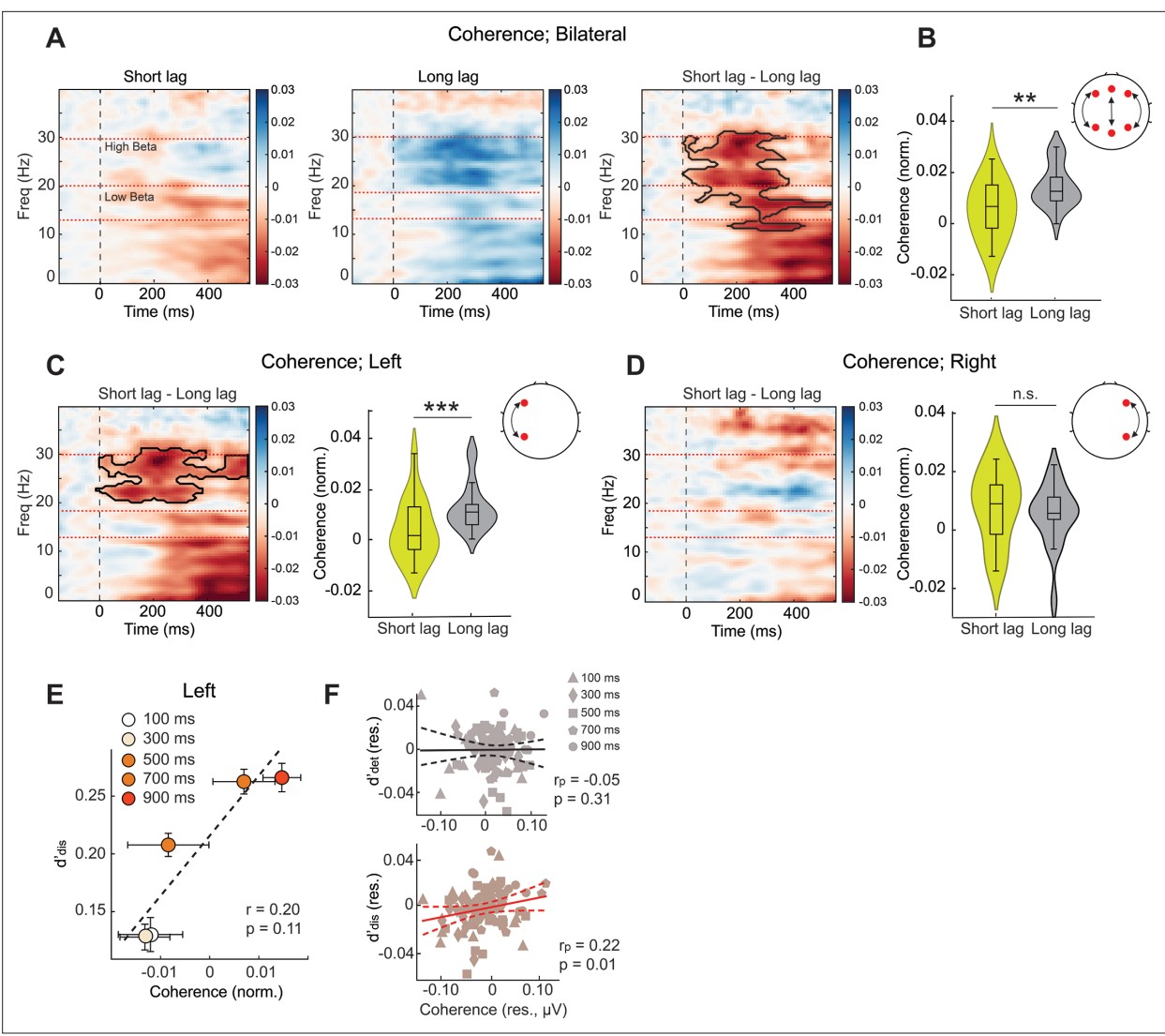

**Figure 5.** High-beta fronto-parietal coherence signals discrimination sensitivity deficits. (**A**). Coherence between frontal and parietal electrodes (Materials and methods) as a function of time (x-axis) and frequency (y-axis) locked to T2 onset (t=0) (n=18 participants), normalized frequency-wise by a baseline mean (gray horizontal bar), and computed by subtracting the average coherograms for correct rejection trials. Cooler colors: higher coherence. Dashed horizontal line: low-beta (13–19 Hz) and high-beta (20–30 Hz) sub-bands. (Left and middle) Coherograms for the short (100+300ms) and long (700+900ms) lag trials, respectively. (Right) Difference coherogram. Bold black outline: statistically significant coherence difference between short and long lag trials (cluster forming threshold p<0.05). (**B**) Bilateral fronto-parietal coherence values in the high-beta band across participants, separately for the short (100+300ms, green) and long (700+900ms, gray) lag trials. Topoplot: schematic of electrode pairs (red circles and black arrows) used for computing bilateral frontoparietal coherence (Materials and methods). Other conventions are the same as in *Figure 3G*. (**C**) (Left) Same as in panel A (right panel) but showing the difference coherogram for the left hemispheric frontoparietal electrodes. Other conventions are the same as in panel A. (Right) Same as in panel B but showing high-beta coherence values over the left hemispheric fronto-parietal electrodes. Other conventions are the same as in panel B and *Figure 3G*. (**D**) (Left and right) Same as in panel C but showing the difference coherogram for the right hemispheric fronto-parietal electrodes. Other conventions are the same as in panel C. (**B–D**) Asterisks: *p<0.05, **p<0.01, and ***p<0.001. (**E**) Same as *Figure 4E* but showing the variation of discrimination sensitivity (d'$_{dis}$, y-axis) with left fronto-parietal high-beta coherence (x-axis) across inter-target lags. Other conventions are the same as in *Figure 4E*. (**F**) Same as in *Figure 4F* but showing the partial correlation of high-beta left fronto-parietal coherence with d'$_{det}$ while controlling for d'$_{dis}$ (top), or, conversely, with d'$_{dis}$ while controlling for d'$_{det}$ (bottom). Other conventions are the same as in *Figure 4F*.

decreased significantly during the blink (short lag-long lag: $\Delta\|\eta_{det}\|$=−1.30 ± 0.70, z=−3.68, p=0.006, BF = 20; $\Delta\|\eta_{dis}\|$=−1.23 ± 0.42, z=−3.54, p<0.001, BF >10$^2$; *Figure 6C–D*). In other words, neural representations encoding the presence of the target, as well as those encoding the target's features, became more overlapping – or less differentiated – during the attentional blink.

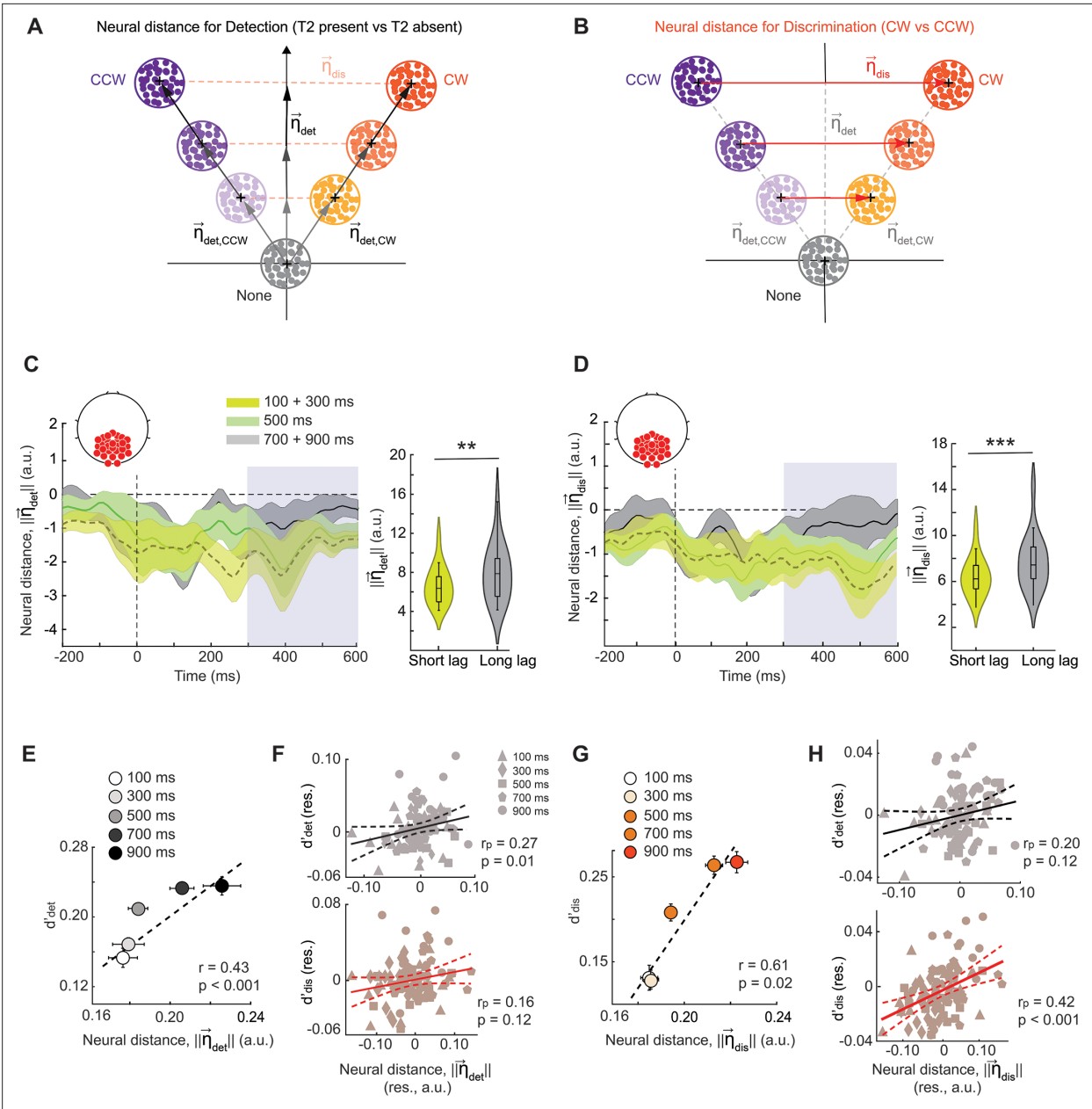

**Figure 6.** Distinct neural dimensions encode detection and discrimination bottlenecks. (**A**) Schematic showing the 'detection' dimension, η_det (y-axis): a linear dimension in a multidimensional electrode space along which neural activity encodes the presence versus absence of T2 (Methods). Gray, orange, and purple dot clusters: distribution of EEG activity for T2 absent, T2 clockwise (CW), and counterclockwise (CCW) trials, respectively. Darker (lighter) shades denote longer (shorter) inter-target lags. (**B**) Same as in panel A but showing the 'discrimination' dimension, η_dis (x-axis): a linear dimension along which neural activity encodes the orientation of T2 (clockwise versus counterclockwise of vertical). (**C**) (Left) Time evolution of the average inter-class distance along the detection dimension ($\|\vec{\eta}_{det}\|$) locked to T2 onset (n=18 participants). Values at each time point are plotted relative to the longest lag (900ms). Bright green, dark green, and black traces: short (100+300ms), intermediate (500ms), and long (700ms) lags, respectively. Shading: s.e.m. Dashed vertical line: T2 onset. Gray shading: Time epoch for neural distance quantification and statistical comparison. (Right) Distribution of $\eta_{det}$ inter-class distances across participants separately for the short (green, 100+300ms) and the long (gray, 700+900ms) lag trials. Asterisks: *p<0.05, **p<0.01, and ***p<0.001. Other conventions are the same as in **Figure 3G**. (**D**) Same as in panel C but showing the time evolution of the average neural inter-class distance along the discrimination dimension ($\|\vec{\eta}_{dis}\|$) (left), and the distribution of neural distances along the discrimination dimension (right). Other conventions are the same as in panel C. (**E**) Same as in **Figure 4C**, but showing variation of detection sensitivity (d'_det, y-axis) with neural distance along the detection dimension ($\|\vec{\eta}_{det}\|$, x-axis) across inter-target lags. (**F**) Same as in **Figure 4D** but showing the partial correlation of $\|\vec{\eta}_{det}\|$ with d'_det while controlling for d'_dis (top), or, conversely, with d'_dis while controlling for d'_det (bottom). (**G**) Same as in **Figure 4E**, but showing variation of discrimination sensitivity (d'_dis, y-axis) with neural distance along the discrimination dimension ($\|\vec{\eta}_{dis}\|$, x-axis) across inter-target lags. (**H**) Same as in **Figure 4F** but showing the partial correlation of $\|\vec{\eta}_{dis}\|$ with d'_det while controlling for d'_dis (top), or, conversely, with d'_dis while controlling for d'_det (bottom).

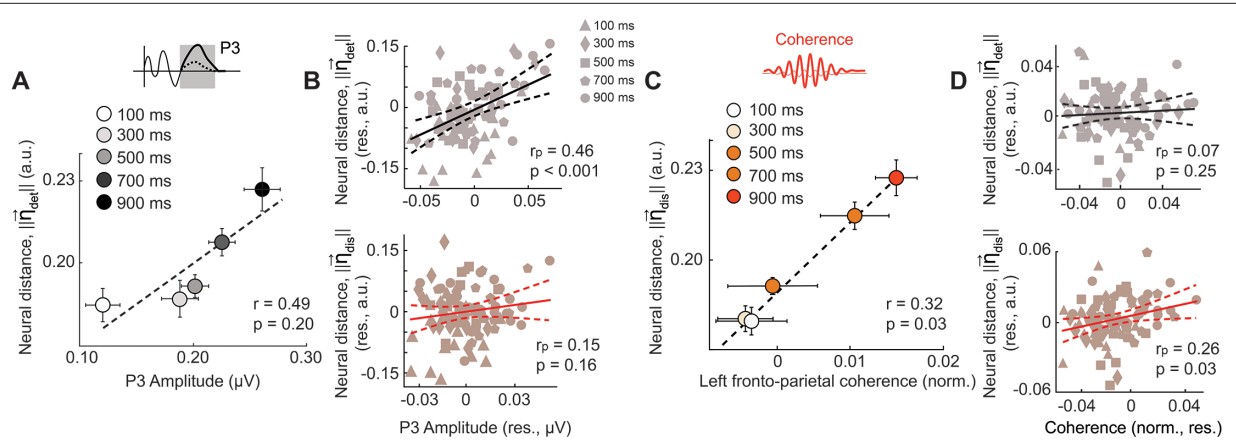

**Figure 7.** Neural dimensions of detection and discrimination deficits correlate with their respective EEG markers. (**A**) Same as in **Figure 4C**, but showing variation of the neural distance along the detection dimension (||η_det||, y-axis) with P3 amplitude (d'_det, x-axis) across inter-target lags. (**B**) Same as in **Figure 4D** but showing the partial correlation of the P3 amplitude with ||η_det|| while controlling for ||η_dis|| (top), or, conversely, with ||η_dis|| while controlling for ||η_det|| (bottom). (**A–B**) Other conventions are the same as in **Figure 4C–D**. (**C**) Same as in **Figure 4E**, but showing variation of the neural distance along the discrimination dimension (||η_dis||, y-axis) with left high beta frontoparietal coherence (x-axis) across the 5 inter-target lags. (**D**) Same as in **Figure 4F** but showing the partial correlation of the left high beta frontoparietal coherence with ||η_det|| while controlling for ||η_dis|| (top), or, conversely, with ||η_dis|| while controlling for ||η_det|| (bottom). (**C–D**) Other conventions are the same as in **Figure 4E–F**.

The online version of this article includes the following figure supplement(s) for figure 7:

**Figure supplement 1.** Linking neural dimensions with neural markers of detection deficits – N2p peak amplitude.

We tested whether the blink-induced collapse along the detection and discrimination neural dimensions would have distinct consequences for behavior. Neural inter-class distance along the detection dimension, $\| \eta_{det} \|$, revealed a significant partial correlation with detection d' across lags ($r_p = 0.27$, p=0.018), but was not correlated with discrimination d' ($r_p = 0.16$; p=0.122; **Figure 6E–F**). Conversely, neural inter-class distance along the discrimination dimension, $\| \eta_{dis} \|$, was significantly correlated with discrimination sensitivity ($r_p = 0.42$, p<0.001) but not with detection sensitivity ($r_p = 0.20$, p=0.122; **Figure 6G–H**). These results suggest a clear, double dissociation between the neural dimensions that underlie the detection and discrimination deficits induced by the attentional blink.

Next, we tested if inter-class distances along the detection and discrimination dimensions correlated with the magnitude of blink-associated ERPs – the N2p or the P3. The amplitude of the occipital N2p and parietal P3 was both significantly correlated with detection distance, $\| \eta_{det} \|$ (N2p: $r_p = 0.31$, p=0.020; P3: $r_p = 0.46$, p<0.001; **Figure 7A–B**, **Figure 7—figure supplement 1A**), but not with discrimination distance, $\| \eta_{dis} \|$ (N2p: $r_p = 0.07$, p=0.330; P3: $r_p = 0.15$, p=0.160). These results further confirm that the N2p and P3 ERP components indicate detection, but not discrimination, bottlenecks underlying the attentional blink.

Finally, we tested whether long-range synchronization between the fronto-parietal cortex also varied with distances along these neural dimensions. In particular, we tested if left fronto-parietal coherence correlated with distances along either the detection or the discrimination neural dimension ($\eta_{det}$ or $\eta_{dis}$, respectively). In line with our earlier findings, left frontoparietal coherence in the high-beta band correlated significantly and specifically with discrimination distance, $\| \eta_{dis} \|$ ($r_p = 0.26$, p=0.039; **Figure 7C–D**), but not with detection distance, $\| \eta_{det} \|$ ($r_p = 0.07$, p=0.255). No such correlation occurred for either right frontoparietal coherence or for coherence in the low-beta band (**Supplementary file 1D**). In other words, left hemispheric fronto-parietal synchronization in the high-beta band provides a key marker for the discrimination, but not detection, bottleneck underlying the attentional blink.

Taken together, these results indicate that the attentional blink impaired sensitivity both for detecting T2 and discriminating its features, leaving criteria unaffected. Yet, neural markers signaling detection and discrimination deficits were distinct. Detection deficits were signaled by specific ERP components, namely the occipitoparietal N2p and the parietal P3. On the other hand, discrimination deficits were signaled by long-range high-beta synchronization between left hemispheric frontal

and parietal cortex. Additionally, distinct neural dimensions ($\eta_{\text{det}}$ and $\eta_{\text{dis}}$) evidenced representational collapse associated with detection and discrimination deficits and correlated systematically with neural markers of each type of deficit (ERPs and beta coherence, respectively). These results reveal a clear dissociation between the behavioral and neural bases of detection and discrimination bottlenecks underlying the attentional blink.

## Discussion

When two successive targets appear in close temporal proximity, behavioral judgments about the second target are compromised: a phenomenon termed the attentional blink (*Raymond et al., 1992*). We show that the attentional blink induces dissociable bottlenecks with both detecting the second target and discriminating its features. Each of these bottlenecks affects the fidelity of the second target's perceptual representations, but neither affects downstream criteria for decisions. Moreover, these bottlenecks map onto distinct neural markers, indicating putatively dissociable mechanisms mediating detection and discrimination deficits.

It is increasingly clear that detection and discrimination are separable processes, each mediated by distinct neural mechanisms. Behaviorally, accurately identifying the first target, versus merely detecting it, produces stronger deficits with identifying the second target (*Broadbent and Broadbent, 1987*). Moreover, dissociable mechanisms have been reported to mediate object detection and discrimination in visual adaptation contexts (*Hillis and Brainard, 2007*). Neurally, shape detection and identification judgements produce activations in non-overlapping clusters in various brain regions in the visual cortex, inferior parietal cortex, and the medial frontal lobe (*Straube and Fahle, 2011*). Similarly, occipital ERPs associated with conscious awareness also show clear differences between detection and discrimination. For instance, an early posterior negative component (200–300ms) was significantly modulated in amplitude by success in detection, but not in identification (*Koivisto et al., 2017*). The closely related visual awareness negativity (VAN) was substantially stronger at the detection, compared to the discrimination, threshold (*Wiens et al., 2023*).

Furthermore, a significant body of previous work has reported dissociable behavioral and neural mechanisms underlying attention's effects on target detection versus discrimination. Behavioral studies have reported distinct effects on target detection versus discrimination in both endogenous (*Correa et al., 2004*) and exogenous (*Lubbe et al., 2005*) attention tasks. Neurally, improved detection of attended targets is accompanied by enhanced neuronal firing (*Treue and Maunsell, 1996*) or higher ERP amplitudes (*Luck et al., 1994*). On the other hand, improved discrimination of target features may be achieved by selectively facilitating the activity of feature-selective neurons (*McAdams and Maunsell, 1999*), especially those that carry maximally discriminative information about the target's features (*Navalpakkam and Itti, 2007*). In addition, attention-induced decorrelation of neuronal firing is hypothesized to systematically aid stimulus discriminability (*Kanashiro et al., 2017*), especially under conditions in which signal correlations are high among pairs of neurons (*Averbeck et al., 2006*). Furthermore, cortical network states become more distinct across object representations, potentially facilitating target discrimination in endogenous attention tasks (*Rotermund et al., 2009*). Hence, there was no reason to expect, a priori, that impairments induced by the attentional blink on target detection and discrimination would be correlated.

In line with this hypothesis, we discovered that the attentional blink induced dissociable detection and discrimination deficits. There was no statistically significant correlation between these two types of deficits within and across participants, and evidence for such a correlation was weak, at best. Unlike previous target identification designs that conflated attentional blink's effect on detection versus discrimination performance (*Raymond et al., 1992*; *Nieuwenstein et al., 2009*; *Chun and Potter, 1995*; *Vogel and Luck, 2002*; *Gross et al., 2004*), our 3-AFC task, and associated signal detection model enabled quantifying each of these deficits separately and identifying a double dissociation between their respective neural correlates. Our dissociation of the attentional blink into distinct subcomponents is complementary to recent studies, which examined whether the attentional blink reflects an all-or-none phenomenon (*Sy et al., 2021*; *Karabay et al., 2022*). For example, the T2 deficit induced by the attentional blink can be either all-or-none or graded, depending on whether T1 and T2 judgements involve distinct or common features, respectively (*Sy et al., 2021*). While a graded change in precision could reflect sensitivity effects, an all-or-none change in guess rates – without a concomitant change in precision – may reflect a criterion increase (conservative detection bias) effect.

Future experiments that incorporate a three-alternative response, with concurrent detection and discrimination, along with key task elements of these earlier studies, may further help resolve these findings. The occipital N2p and parietal P3 amplitudes were modulated strongly by the attentional blink and correlated systematically and selectively with detection deficits. Conversely, left hemispheric fronto-parietal high-beta coherence was, again, modulated significantly by the attentional blink and strongly predictive, specifically, of discrimination deficits. By contrast, neither the early occipital P1 nor the frontocentral P2 systematically varied with either detection or discrimination deficits, although the latter ERP was significantly diminished during the attentional blink.

Interestingly, we found no evidence indicating that these two computations (detection and discrimination) were sequential; in fact, the modulation of beta coherence occurred almost immediately after T2 onset and lasted well afterwards (>400ms from T2 onset; *Figure 5A–B*), suggesting that an analysis of T2's features proceeded in parallel with its detection and consolidation. We also modeled detection and discrimination as concurrent computations in our SDT model (*Figure 3A–B*). Previous work suggests that while object detection and categorization processes proceed in parallel, detection and identification processes occur sequentially (*Grill-Spector and Kanwisher, 2005*). Our results are in line with this literature, if we consider T2's discrimination judgement – clockwise versus counter-clockwise of vertical – to be a categorization, rather than an identification judgement. Moreover, this earlier study (*Grill-Spector and Kanwisher, 2005*) observed significant trial-wise correlations between detection and categorization responses, suggesting that the two processes involve the operation of the same perceptual filters ('analyzers'). Our study, on the other hand, reports distinct neural bases for detection and discrimination computations. Yet, the two sets of findings are not mutually contradictory.

In many conventional attentional blink tasks (*Raymond et al., 1992*; *Sergent et al., 2005*; *Vogel and Luck, 2002*), complex visual stimuli, like letters, must be detected among a stream of background distractors with closely similar features, such as digits. In this case, target detection would require the operation of shape-selective perceptual filters for feature analysis. These same shape-selective filters would be involved also for discriminating between distinct, but related target stimuli (e.g. two designated candidate letters). In our task, target gratings needed to be distinguished in a stream of plainly distinct background distractors (plaids), whereas the discrimination judgement involved analysis of grating orientation. As a result, our task design likely precludes the need for the same perceptual filters in the detection and the discrimination judgements. Absent this common feature analysis, our results suggest distinct electrophysiological correlates for the detection and discrimination of targets.

These results extend, and further advance, a significant literature on the attentional blink effects on ERPs (*Vogel et al., 1998*; *Dell'Acqua et al., 2006*; *Sergent et al., 2005*; *Lasaponara et al., 2015*; *Vogel and Luck, 2002*; *Kranczioch et al., 2003*; *Zivony and Lamy, 2022*; *Akyürek et al., 2010*; *Jolicœur et al., 2006*; *Martens et al., 2006*). Several previous studies have shown that the occipital N2 component exhibits a higher amplitude on short lag trials in which T2 was successfully detected, versus not (*Vogel et al., 1998*; *Sergent et al., 2005*; *Lasaponara et al., 2015*; *Kranczioch et al., 2003*). Other studies have shown that the attentional blink also affects the amplitude of N2pc, a posterior, contralateralized N2 component (*Akyürek et al., 2010*; *Jolicœur et al., 2006*; *Jolicoeur et al., 2006*); a decrease in the amplitude of the N2pc has been hypothesized to index impaired attentional gating of T2. Similarly, previous studies have shown either partial or complete suppression of P2 during the attentional blink (*Vogel et al., 1998*; *Vogel and Luck, 2002*; *Akyürek et al., 2010*). Yet, these studies reported no change in P2 amplitude between successfully and unsuccessfully detected targets, in line with our own findings. Taken together with our own findings, these results establish the N2 as a key marker for successful detection of T2, achieved by robust attentional gating and filtering out of irrelevant temporal (plaid) distractors.

The effect of the attentional blink on the P3 component has been consistently observed in many studies, and its functional significance has been thoroughly investigated. Many studies have reported a systematic reduction in P3 amplitude and latency in both detection and identification tasks (*Vogel et al., 1998*; *Sergent et al., 2005*; *Lasaponara et al., 2015*; *Kranczioch et al., 2003*; *Akyürek et al., 2010*; *Jolicoeur et al., 2006*). For centrally presented stimuli, the P3 recorded over parietal cortex is generally linked to high-level cognitive functions, such as consolidation into working memory (*Reed et al., 2022*). In the present study, we observed that the parietal P3 amplitude was correlated selectively with detection, rather than discrimination deficits. This suggests that the P3 deficit indexes a specific bottleneck with encoding and consolidating T2 into working memory, rather than an inability

to reliably maintain its features. In this regard, a recent study (*Dellert et al., 2022*) measured ERP correlates of the perceptual awareness of the T2 stimulus whose relevance was uncertain at the time of its presentation. In contrast to earlier work, this study observed no change in P3b amplitude across seen (detected) and unseen targets. Taken together with this study, our findings suggest that rather than indexing visual awareness, the P3 may index detection, but only when information about the second target, or a decision about its appearance, needs to be maintained in working memory. Additional experiments, involving targets of uncertain relevance, along with our behavioral analysis framework, may help further evaluate this hypothesis.

In contrast to the detection deficits associated with ERP amplitude suppression, discrimination deficits were clearly correlated with disrupted high beta coherence in the frontoparietal electrodes. Although previous studies have shown clear modulations in beta-band coherence during the blink epoch (*Kranczioch et al., 2007*; *Gross et al., 2004*), the tasks employed in these studies (e.g. letter identification) could not distinguish between detection and discrimination deficits. To our knowledge, ours is the first study to show that modulation of frontoparietal beta synchrony correlates specifically with deficits associated with discrimination performance. This finding is in line with previous literature, which suggests that frontoparietal beta synchrony plays a key role in visual categorization, a function that relies critically on object feature analysis (*Freedman et al., 2001*). Beta-band synchrony between the PFC and PPC may carry information about task-relevant categories (*Antzoulatos and Miller, 2016*): mechanistically, the oscillations may serve to selectively communicate task-relevant features between PFC and PPC (*Antzoulatos and Miller, 2016*). As a result, impairments in beta synchrony may hinder the selection of relevant features leading to a selective discrimination deficit but not a detection deficit, as observed in our task. Moreover, previous studies have reported a disruption in beta-band coherence between the left frontal and right parietal cortex (*Kranczioch et al., 2007*; *Gross et al., 2004*). Yet, we observed significant left lateralization of frontoparietal beta-band coherence during the attentional blink (*Figure 5B*). One possibility is that this lateralization arises from beta oscillations in the left motor cortex signaling right-handed responses (*Murthy and Fetz, 1992*). We discount this hypothesis for multiple reasons. First, the left-lateralization of the high-beta coherence occurred closely following T2 onset, many hundreds of milliseconds before the response epoch. Second, the beta coherence deficit correlated specifically with only one kind of perceptual deficit (discrimination, but not detection). Finally, in our tasks, participants reported T2's orientation with a bimanual response – the left index finger to report counter-clockwise orientations and the right index finger to report clockwise orientations. In other words, motor-activity linked beta oscillations were highly unlikely to be the source of the lateralized coherence deficits observed in our study.

Both successful target detection and discrimination improve significantly with stimulus contrast (*Anton-Erxleben and Carrasco, 2013*). We had, therefore, expected to observe stronger detection and discrimination d' deficits for low contrast, as compared to high contrast targets. Surprisingly, we found no significant effect of contrast on either type of deficit (*Figure 2A–B*). In other words, high (100%) contrast T2 stimuli were also strongly susceptible to the detection and discrimination bottlenecks associated with the attentional blink. Thus, despite a clear contrast-dependent encoding of T2 in early sensory cortex, the attentional blink produced a significant deficit with downstream processing, even for targets of high contrast. While at odds with some earlier work, which suggests an early-stage perceptual bottleneck (*Chua, 2005*; *Dux and Marois, 2009*; *Simione et al., 2012*), these results are largely consistent with findings from the majority of previous studies (*Raymond et al., 1992*; *Shapiro et al., 1997*; *Chun and Potter, 1995*; *Di Lollo et al., 2005*; *Luck et al., 1996*; *Sergent et al., 2005*; *Chua, 2005*; *Jolicoeur, 1999*; *Jolicoeur, 1998*) which suggest a late-stage bottleneck.

But what specific neural computation underlies this late bottleneck? Previous research has documented three key computational stages mediating perceptual decisions: encoding of sensory evidence, formation of a decision variable, and application of a decision rule when formulating the choice (*Gold and Shadlen, 2007*). Our results indicate that the attentional blink impacts, specifically, the second stage, associated with transforming T2's sensory evidence into a decision variable. In fact, our results indicate that the attentional blink did not affect the final, decisional stage, at all: criteria, for both detection and discrimination judgments, were essentially identical across inter-target lags.

These results appear at odds with earlier work, suggesting that the attentional blink produced both sensitivity and criterion effects in a detection task (*Lasaponara et al., 2015*; *Caetta and Gorea, 2010*). *Lasaponara et al., 2015* reported a decrease in detection bias (likelihood ratio, or LR bias)

with lag, whereas *Caetta and Gorea, 2010* reported an increase in detection criterion with lag. Our study, on the other hand, reports no change in detection criterion with lag. Our results are potentially consistent with those of Lasaponara et al. Given that d' increases with lag, a constant (positive) detection criterion would mathematically yield a decreasing likelihood ratio bias with lag (LR bias = exp (-d' * c)). In other words, a decrease in LR bias with lag could occur simply because of the increase in detection d' without a concomitant change in detection criterion. On the other hand, a major difference between Caetta et al's study (*Caetta and Gorea, 2010*) and ours is that Caetta et al modeled a fixed (unvarying) detection threshold across inter-target lags, yielding changes in detection criteria; the latter (detection criteria), and not thresholds, are the conventional SDT measure of bias (*Caetta and Gorea, 2010*). Their choice of unvarying thresholds was based on the assumption that participants could not predict the inter-lag of the trial in advance when different lags are interleaved across trials, and, therefore, could not adopt different detection thresholds across lags. By contrast, we considered it more reasonable that participants could modulate their detection thresholds across lags, given that decisions about T2 were made at the end of each trial, at which time the participant would be well aware of the inter-target lag for that trial. To explore this hypothesis further, we tested various models including these two models: one with a fixed detection threshold and the other with a

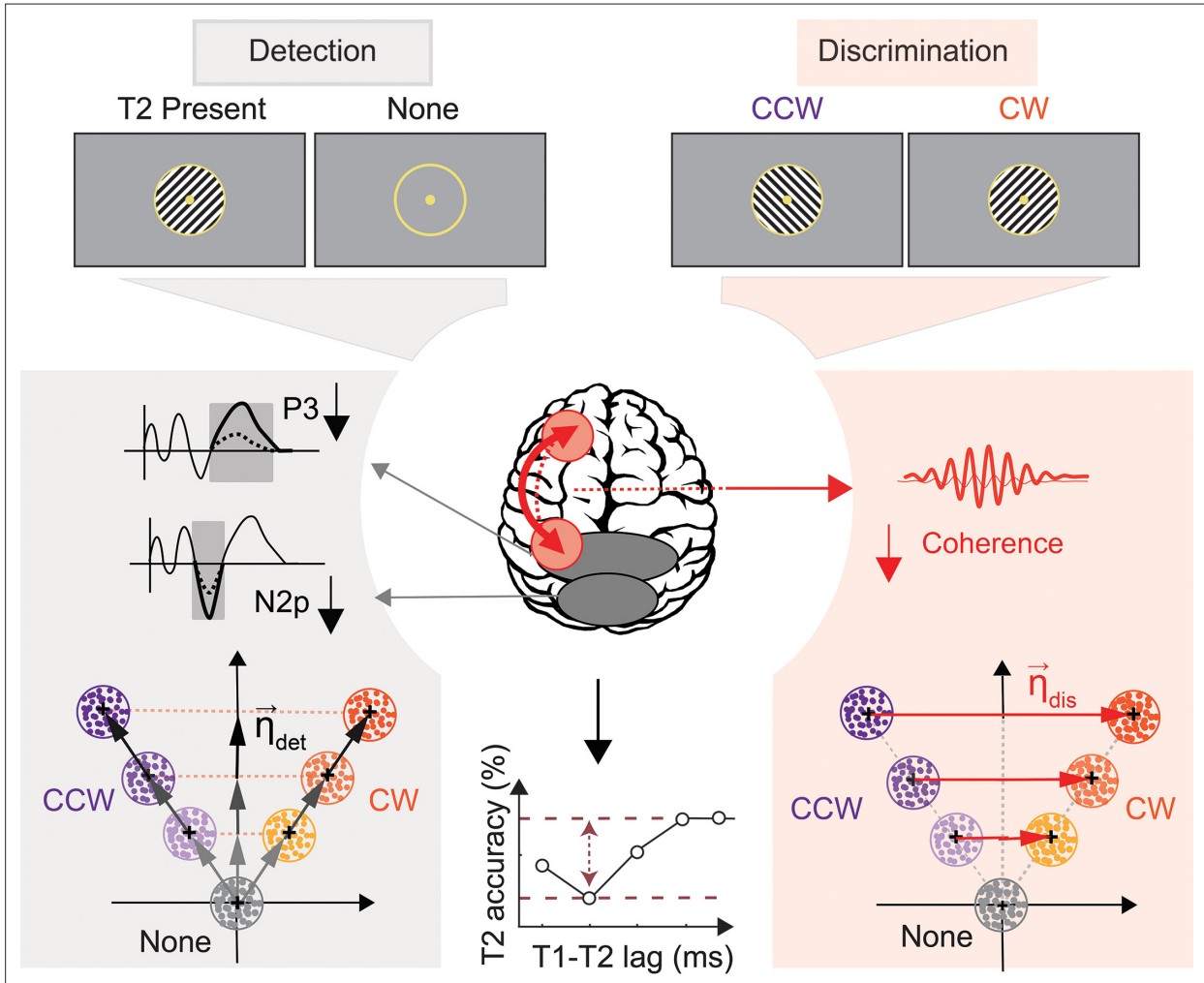

**Figure 8.** Distinct neural bases of subcomponents of the attentional blink. The attentional blink selectively impairs a specific component of attention – perceptual sensitivity (**d'**) – and produces both detection (top, left) and discrimination (top, right) deficits. Detection d' deficits – deficits with distinguishing the presence versus absence of the second target (**T2**) – are correlated with reduced amplitudes of N2p and P3 ERPs (gray shading, left top). They are also accompanied by a representational collapse along the detection dimension (gray shading, left bottom). By contrast, discrimination d' deficits – deficits with discriminating T2's orientation – is evidenced by reduced left fronto-parietal beta coherence (red shading, left top) and a representational collapse along the discrimination dimension (red shading, left bottom).

variable detection threshold across lags. The results clearly supported a model with varying thresholds and confirmed a constant detection criterion (bias) across lags in our data.

Taken together, these results suggest the following testable schema (*Figure 8*). First, smaller N2p and P3 amplitudes may reflect deficits in the activation of neural populations involved in gating and consolidating, respectively, the T2 stimulus into working memory. Second, reduced fronto-parietal high-beta coherence may index deficits in a top-down mechanism that enables discriminating T2's features; the latter computation relies on accurate discrimination of clockwise versus counter-clockwise target orientations. Future research with causal interventions like tACS or TMS could test this mechanistic schema of specific brain regions or oscillatory patterns in mediating distinct aspects of the attentional blink. More broadly, these findings contribute to our understanding of the relationship between attention and perception and may be relevant for designing rational interventions for neurological disorders that produce attention deficits.

## Materials and methods
### Participants
Twenty-four healthy individuals (9 females; age range: 20–28 years) participated in the study. All participants had normal or corrected to normal vision with no known history of neurological disorders. Among these, scalp EEG recordings were acquired concurrently with behavior for eighteen participants. Experimental procedures were approved by the Institute Human Ethics Committee, Indian Institute of Science, Bangalore. Participants provided informed, written consent prior to participating in the experiments.

### Behavioral data acquisition
#### Apparatus and stimuli
Participants were tested with a typical Attentional Blink (AB task) protocol involving a rapid serial visual presentation (RSVP) stream (frame rate: 10 stimuli per second; 70ms on, 30ms off). Stimulus presentation and data acquisition were programmed with Psychtoolbox (*Kleiner et al., 2007*) and MATLAB 2017b MathWorks Inc (2017), (Natick, MA.). Participants were seated with their head resting on a custom chin rest in a dimly lit room, with their eyes positioned 60 cm away from a contrast-calibrated visual display (24″ LCD monitor, 100 Hz refresh rate, BenQ Corp.). Responses were recorded with an RB-840 response box (Cedrus Inc). Participants were instructed to maintain gaze fixation at a central dot throughout the experiment and to avoid eye blinking during the stimulus presentation. Fixation, eye blinks, and eye movements were monitored monocularly at 1000 Hz, with an infrared-based eye tracker (Eyelink, SR Research).

#### Task procedure
Each trial commenced with gaze fixation on a central dot (diameter: 0.5° of visual angle/dva, color: yellow) in the middle of a gray (35 cd/m$^2$ maximum luminance) screen. After 500ms, task-irrelevant distractors – full contrast plaids (diameter: 4 dva) – appeared in the center of the screen in an RSVP stream (100ms per stimulus). The plaid comprised of two superposed, square wave gratings (spatial frequency: 1.8 cpd) oriented at 45° and 135° relative to the vertical, displayed within a circular mask (diameter: 4 dva). The phase of the plaid stimuli underwent a 180° phase inversion (full white to full black, and vice versa) across successive frames of the RSVP presentation. Plaids were encircled by a circular placeholder (diameter: 4.1 dva, thickness: 0.1 dva). The fixation dot and the placeholder were present on the screen throughout the trial.

Following a variable interval (300–1300ms, geometrically distributed) of plaid presentation, the first target stimulus (T1, oriented square-wave grating, spatial frequency: 0.6 cpd) appeared in the RSVP stream (*Figure 1A*). Following another variable interval, the second target stimulus (T2, oriented square-wave grating, 1.8 cpd), higher spatial frequency than T1, was presented in the RSVP stream. The interval between T1 and T2 onsets was geometrically distributed and pseudorandomly selected from one of five possible 'inter-target' lags (100, 300, 500, 700, or 900ms); plaids, identical with those described previously, were presented in the interleaving frames. The T2 grating appeared on only two-thirds of the trials ('T2 present' trials). On the remaining third of the trials, randomly interleaved with the T2-present trials, no T2 was presented ('no T2' trials); rather, the plaid was replaced

with a gray filled circle (0% contrast) inside the circular placeholder. In addition, on 50% of the trials, T2 gratings were presented at full contrast, whereas on the remaining 50% of the trials (randomly interleaved), T2 gratings were presented at a contrast staircased for each participant individually (see next). T1 orientations, as well as T2 orientations and contrasts, were proportionately counterbalanced across all inter-target lags. Finally, plaids were presented for a set of six frames (600ms) and participants were probed for response.

At the end of each trial, participants provided two responses, one for each of T1 and T2, in succession (*Figure 1A*). First, participants indicated whether the T1 grating's orientation was closer to the cardinal (0°, 90°) or to the diagonal (45°, 135°) axes with one of two button presses (2-alternative forced choice/2-AFC, *Figure 1A*). Second, participants indicated whether the T2 grating's orientation was clockwise of vertical (CW), counter-clockwise of vertical (CCW), or if it was not detected at all (None), with one of three distinct button presses (3-AFC, *Figure 1A*).

## Training and testing

Two behavioral sessions were conducted on two consecutive days. The first session (on Day 1) comprised of training (30 min) followed by a staircasing session (~1.5 hr). The training session typically comprised 2–4 blocks of trials with 84 trials each. Only the longest inter-target lags (700 and 900ms) were included; short-lag trials were excluded both from this and the subsequent staircasing sessions based on previous reports, which suggest that practice effects can mitigate the magnitude of the attentional blink at short lags (*Nakatani et al., 2012*). At the end of each trial, participants received on-screen feedback regarding their success or failure for each target. For this session, T1 and T2 were both presented at full contrast; T1 was presented with a ±2° deviation from cardinal or diagonal orientations, pseudo-randomly counterbalanced among the 4 different options, whereas T2 was presented at ± 10° deviation from the vertical. Note that smaller (larger) angular deviations render the cardinal/diagonal (CW/CCW) judgement with the T1 (T2) grating easier. Following this, participants performed a second, staircasing session (~400 trials). In this session, T1's angular deviation from the cardinal/diagonal orientations was staircased to achieve 87.5% discrimination accuracy for T1, whereas T2's contrast was concurrently staircased to achieve 75% discrimination accuracy for T2. T1's angular deviation and T2's contrast determined from this session were used for the main experiment on Day 2; while the former angle was used for all T1 trials, the latter contrast was employed in the low-contrast T2 trials alone.

On Day 2, participants were re-familiarized with the task for ~150–200 trials before the main experiment commenced. Before testing, participants were provided with a 30-min break. The main experiment comprised trials with all 5 inter-target lags (see above) and comprised 3 blocks of 312 trials each (total of 936 trials per participant). No feedback was provided after each trial. Each block was subdivided into 4 mini-blocks of 78 trials, and participants were provided short breaks (~5 min) after each mini-block. Data from the staircase and training sessions were excluded from subsequent analyses.

## Eye-tracking and data exclusion

Throughout each experiment, participants' fixation was monitored and gaze position was recorded and stored for offline analysis. Trials in which an eye blink occurred any time before the response period were excluded from the final analyses. In addition, trials in which the gaze positions deviated from the central pedestal ±50ms before or after the presentation of either the T1 or the T2 gratings (radius: 2 dva from fixation dot) were excluded from further analysis. Following these exclusions, the median trial rejection rate was 2.1% [0.8–3.4%] (median, 95% confidence interval) across participants.

## Behavioral data analysis

### Measuring attentional blink effects on psychometric quantities

We quantified the effect of the attentional blink on behavioral detection and discrimination psychometrics for the T2 grating. All of the analyses were performed including only trials in which participants made correct responses for the T1 judgment. Our 3-alternative task for T2 judgements, and the associated behavioral model (see next), enabled quantifying psychophysical parameters linked to both detection and discrimination within a single task and analysis framework. Participants' 3-AFC responses for the T2 grating were collated into five 3x3 stimulus-response contingency tables, separately for each of the five inter-target lags and each T2 contrast (high, low). The rows of each

contingency table represent the three stimulus events (T2 CW, T2 CCW or no T2), and the columns represent the participants' choices among the three alternatives (*Figure 2C*). Each table comprises five response types – (i) Hits (H, elements along the main diagonal, except the bottom right) wherein the subject correctly identifies the orientation of T2 as CW or CCW, (ii) Misses (M, last column, except the bottom right), wherein the subject reported 'no T2' on trials in which T2 was actually presented, (iii) False alarms (FA, last row, except the bottom right), wherein the subject incorrectly reported the presence of T2 (as CW or CCW) when no T2 was presented, (iv) Correct Rejections (CR, bottom right), wherein the subject accurately reported the absence of T2, and (v) Misidentifications (MI, all other entries), wherein the subject incorrectly reported T2's orientation (CW grating as CCW, or vice versa).

Based on these psychometric measures, we computed detection and discrimination accuracies as follows. Detection accuracies were computed as the average proportion of the hits, misidentification, and correct rejection responses; misidentifications were included because not missing the target reflected accurate detection. By contrast, discrimination accuracies were computed based on the average proportion of the two correct identifications (hits) on T2 present trials alone. We performed two-way ANOVAs on both detection and discrimination accuracies with the inter-target lag (5 values) and T2 contrast independent factors. We found main effects of both lag ($F_{(4,92)}=18.81$, $p<0.001$) and contrast ($F_{(1,92)}=21.78$, $p<0.001$) on detection accuracy, but no interaction effect between lag and contrast ($F_{(4,92)}=1.92$, $p=0.113$). Similarly, we found main effects of both lag ($F_{(4,92)}=25.08$, $p<0.001$) and contrast ($F_{(1,92)}=16.58$, $p<0.001$) on discrimination accuracy, but no interaction effect between lag and contrast ($F_{(4,92)}=0.93$, $p=0.450$). Post-hoc tests based on Tukey's HSD revealed a significant difference in discrimination accuracies between the two shortest lags (100ms and 300ms) and the two longest lags (700 and 900ms) for both low and high contrast targets, and for both detection and discrimination accuracies ($p<0.01$). But they revealed no significant difference between the two shortest lags ($p>0.25$) or the two longest lags ($p>0.40$) for either target contrast or for either accuracy type. As a result, for subsequent analyses, we pooled together the 'short lag' (100ms and 300ms) and the 'long lag' (700ms and 900ms) trials. We quantified the effect of the attentional blink on each of the psychometric measures as well as detection and discrimination accuracies by comparing their respective average values between the short lag and long lag trials, separately for the high and low T2 contrasts.

## Measuring attentional blink effects on psychophysical parameters

We estimated psychophysical parameters by fitting the 3-AFC contingency tables, described above, with a novel signal detection theory (SDT) model. Observers' decisions were modeled in a two-dimensional decision space to fit T2 detection and discrimination performance simultaneously; the model was inspired by the recently published m-ADC model (*Sridharan et al., 2014*) widely used for the analysis of multialternative attention tasks.

### Signal detection model description

On each trial, T2 stimulus' features were encoded with a bivariate Gaussian decision variable ($\psi$). The x-axis of this decision space represents evidence for T2's orientation (CW or CCW), whereas the y-axis represents evidence for T2's presence. In other words, a larger signal magnitude along the x-axis (either positive or negative) favors better discrimination, whereas a higher signal magnitude along the y-axis favors more reliable detection. The mean of the bivariate decision variable's value along the x-axis and y-axis is parameterized by two perceptual parameters, indexing perceptual discriminability ($\pm d'_{dis}$) for discriminating T2's orientation, and perceptual sensitivity ($d'_{det}$) for detecting T2's presence, respectively. On the T2 absent trials, $\psi$ is drawn from a distribution centered at the origin ($d'_{dis} = d'_{det}=0$). On T2 present trials, the decision variable $\psi$ is drawn from one of four distributions (*Figure 3A*), depending on T2's contrast (*Figure 3A*; high, low) or its orientation (*Figure 3A*; CW, CCW).

The participant makes one of three choices – T2 CW, T2 CCW, or no T2 – based on a Y-shaped decision surface (*Figure 3B*) that divides the decision space into three zones (red, blue, or gray-shaded regions, respectively). This decision surface is parameterized by three decisional parameters: (i) a detection criterion ($c_{det}$) that determines the displacement of this surface along the vertical direction (y-axis), (ii) a discrimination criterion ($c_{dis}$) that determines the displacement of this surface along the horizontal direction (x-axis) and (iii) an angle ($\beta$) between the two arms of the Y (*Figure 3B*).

The subject's decision on each trial is based on the decision zone into which $\psi$ fell on that trial. Note that the proportion of each type of response for each stimulus event type CW, CCW, or no T2 in the stimulus-response contingency table can be mapped to the integral of the conditional density of the decision variable within the decision zone for the respective T2 event and response type. These integrals were evaluated numerically (*integral2* function in Matlab).

Parameters (d', c, β) were estimated with a maximum likelihood estimation approach (**Sridharan et al., 2014**). Typically, parameters were fit for each lag, independently, unless specified otherwise (see next). Goodness of fit was assessed using a randomization test based on the chi-squared statistic (**Sridharan et al., 2014**); p<0.05 was taken as evidence of significant deviation of model fits from experimental data.

## Model comparison analysis

We fit observers' response proportions to five models, each with a set of progressively stronger constraints (decreasing model complexity).

### Model I

This model was the least constrained and comprised 35 parameters. 20 perceptual parameters $d'_{det}$ and $d'_{dis}$ were estimated individually for five lags and separately for two T2 contrasts. Similarly, 15 decisional parameters $c_{det}$, $c_{dis}$, and β were estimated individually for each of the 5 lags, separately (15 c and β parameters); because the high and low T2 contrast trials were randomly interleaved within each block, with no a priori information about which type of target contrast would occur, we modeled these decisional parameters as being identical across high and low T2 contrast trials.

### Model II

This model was identical to Model I except that β was constrained to be identical across lags (20 d', 10 c, 1 β). Therefore, this model comprised 31 parameters.

### Model III

This model was identical with Model II, except that additional constraints were imposed on discrimination-related parameters: (i) $d'_{dis}$ was constrained to be identical across contrasts and (ii) $c_{dis}$ was constrained to be identical across lags. This model comprised 22 parameters.

### Model IV

This model was also identical with Model II, except that additional constraints were imposed on detection-related parameters: (i) $d'_{det}$ was constrained to be identical across contrasts and (ii) $c_{det}$ was constrained to be identical across lags. This model also comprised 22 parameters.

### Model V

This model was the most constrained of all and comprised all constraints in Models II-IV. This model comprised 13 parameters.

Formal model comparison was performed with the Akaike Information Criterion (AIC) and Bayesian Information Criterion (BIC; **Vrieze, 2012**). These scores represent a trade-off between model complexity (the number of model parameters) and goodness-of-fit (based on the log-likelihood function); a lower AIC/BIC score represents a better candidate model (**Vrieze, 2012**). Based on this analysis, the best model was determined to be Model 3 (see Results, section on '*Detection and discrimination deficits reflect sensitivity, rather than criterion, effects*').

## Quantifying the attentional blink effect

To quantify the effect of the attentional blink on each of the estimated parameters, the average value of the parameter at the shortest two lags (100 and 300ms, SL) was subtracted from the average value of the parameters at the longest two lags (700 and 900ms, LL). For example, attentional blink effect on detection sensitivity was computed as: $\Delta d'_{det} = d'^{LL}_{det} - d'^{SL}_{det}$. This computation was performed for each T2 contrast separately. In some cases (**Figure 3G**, **Figure 3—figure supplement 2**), the deficit was quantified with a modulation index. For example, MI-$d'_{det} = (d'^{LL}_{det} - d'^{SL}_{det})/ (d'^{LL}_{det} + d'^{SL}_{det})$.

Because, in the best model selected, $c_{dis}$ and $\beta$ were constrained to be equal across lags, no blink effect was calculated for these parameters. Error bars were computed as the SEM across participants.

## EEG data acquisition and preprocessing

### Acquisition

EEG data were recorded with a dense array of scalp electrodes (128 channels, EGI's Geodesic Inc USA), a Net Amps 400 amplifier, and EGI's Net Station software (version 5.2.0.2, Oregon, USA). During acquisition, data were referenced to the Cz electrode and stored for offline analyses. The EEG cap contains a few additional electrodes that are placed around the orbits of the eye and enable monitoring an EOG signal. Channel impedances were monitored and maintained at <50 kΩ during the experiment. Data were sampled at 1000 Hz and stored offline for analysis.

### Preprocessing

The data were pre-processed with functions from the EEGLAB toolbox (v. 14.0.0), Chronux toolbox (v. 2.12.03) as well as custom scripts developed in Matlab. First, the data were downsampled to 250 Hz by decimation and bandpass filtered between 0.5 and 18 Hz (FIR filter, *designfilt* function in Matlab). Next, a bandstop filter from 9 to 11 Hz was applied to remove the 10 Hz oscillations evoked by the RSVP presentation. Channel traces were then re-referenced to the average signal across channels. We then epoched the data into trials and applied SCADS (Statistical Control of Artifacts in Dense Array EEG/MEG Studies *Junghöfer et al., 2000*) to identify bad epochs and artifact-contaminated channels. SCADS detects artifacts based on three measures: maximum amplitude over time, standard deviation over time, and first derivative (gradient) over time. Any electrode or trial exhibiting values outside the specified boundaries for these measures was excluded. The boundaries were defined as $M \pm n * \lambda$, where M is the grand median across electrodes and trials for each of the three measures, and $\lambda$ is the root mean square (RMS) of the deviation of medians across sensors relative to the grand median. We set n to 3, allowing data within three boundaries to be retained. The percentage of electrodes per participant rejected was 6.3 ± 0.43% (mean ± s.e.m. across participants), whereas the percentage of trials rejected per electrode and participant was 3.4 ± 0.33% (mean ± s.e.m.).

Bad channels were linearly interpolated using data from the four nearest neighbor electrodes. Epochs were demeaned based on a baseline computed in the fixation time window (250ms) at the beginning of each trial and then quadratically detrended.

## EEG data analysis

### Event Related Potential (ERP) analysis

ERP estimation for this attentional blink task followed a procedure outlined in *Sergent et al., 2005*. First trials were sorted based on inter-target lags (100, 300, 500, 700, and 900ms). This yielded an average of (200±13, 171±9.71, 145±7.54, 117±5.43, 87±4.51) (mean ± s.e.m. across participants) trials for each of the 5 lags, respectively. Trials from the longest two lags (700 and 900ms) were pooled to provide sufficient trials for robust ERP estimation at these lags. Then, EEG traces were epoched from –300ms before to +700ms after either T1 onset or T2 onset and averaged across trials to estimate T1-evoked and T2-evoked ERPs, respectively. Because of the periodic RSVP presentation, these ERPs included strong, plaid-evoked responses, which occurred time-locked to both T1 and T2. To remove this contribution, we subtracted ERPs associated with T2-absent trials (correct rejection trials only) from ERPs associated with T2-present trials (as in *Vogel et al., 1998*; *Sergent et al., 2005*); this subtraction was performed separately for each inter-target lag. ERP amplitudes were quantified by measuring peak values within specific time windows corresponding to distinct components.

We analyzed five ERP components, routinely studied as neural markers in attentional blink tasks: (i) P1 and N1 components in posterior, lateral electrodes (P7 and P8) between 40–140ms and 90–160ms, respectively, post T2 onset (*Chmielewski et al., 2016*), (ii) N2p component in occipitoparietal electrodes (O1, O2, Oz, P3, P4, Pz and surrounding PO electrodes) between 150 and 300ms post T2 onset (*Dell'Acqua et al., 2006*; *Kranczioch et al., 2007*; *Jolicœur et al., 2006*; *Jolicoeur et al., 2006*), (iii) P2 component in frontocentral electrodes (C3, C4, Cz, F3, F4, Fz) between 150 and 300ms post T2 onset *Vogel et al., 1998*; *Akyürek et al., 2010*, (iv) P3 component in parietal electrodes (P3, P4, Pz) between 300 and 550ms post T2 onset (*Vogel et al., 1998*; *Sergent et al., 2005*; *Lasaponara et al., 2015*; *Kranczioch et al., 2007*). We quantified the change in ERP amplitudes during

the attentional blink (see Materials and methods section on 'Statistical analyses') and also performed partial correlations between the ERP amplitudes and behavioral metrics (see Materials and methods section on 'Partial correlations between neural and behavioral metrics'). All of the key ERP analyses were repeated without the 9–11 Hz bandstop filter and yielded results virtually identical to those reported in the main text and in *Supplementary file 1B*.

## Analysis of long-range synchronization

We also quantified long-range synchronization between the frontal and parietal cortex, an established neural marker of the attentional blink (*Kranczioch et al., 2007*; *Gross et al., 2004*). For this, we quantified the averaged spectral coherence between each pair of electrodes in the frontal (F3, F4, Fz) and parietal (P3, P4, Pz) cortex (9 pairs), respectively. These analyses were performed with the data bandpass filtered between 0.1 and 40 Hz (FIR filter) and without a 9–11 Hz bandstop filter. Coherence measures the degree of functional connectivity at different frequencies between two regions and is quantified as the ratio of the squared cross spectral density between the signals normalized by the product of the individual signals' auto spectral densities.

$$C_{xy}(f) = \frac{|G_{xy}(f)|^2}{|G_{xx}(f)| \cdot |G_{yy}(f)|}$$

where $G_{xy}(f)$ denotes the cross spectral density between signals x and y, whereas $G_{xx}(f)$ and $G_{yy}(f)$ represent their, respective, auto spectral densities. We computed time frequency representations for coherence (a 'coherogram') with the *coherencyc* function in the Matlab Chronux (v2.11) toolbox *Bokil et al., 2010* with a time-half bandwidth product of 3 and 5 tapers (TxW$_{\frac{1}{2}}$=3, K=5). Coherence values were computed in sliding windows of 300ms duration with a 50ms stride computed from –200ms before to +600ms following T2 onset. Coherence values were divisively normalized frequency-wise, on a per-trial basis, with coherence computed in a baseline window from 0 to 300ms prior to T2 onset. As with the ERPs, to remove the contribution of T1-evoked responses, coherence values at each time point and frequency of the T2-absent (correct rejection) trials were subtracted from the coherence values of the T2-present trials; this subtraction was performed for each inter-target lag separately. To quantify the effect of the attentional blink on the coherogram, we subtracted the coherograms estimated from the short lag trials (100 and 300ms) from the corresponding coherograms from the long lag trials (700 and 900ms). Subsequently, we conducted a cluster-based permutation test on this difference coherogram to identify time periods and frequency bands significantly modulated by the attentional blink (cluster forming threshold p<0.05). For subsequent, partial correlation analyses of coherence with behavioral metrics and neural distances (see next), we focused on a 300ms time period (0–300ms following T2 onset) and high-beta frequency band (20–30 Hz) identified by the cluster-based permutation test (*Figure 5A–C*).

## Analysis of neural dimensions corresponding to detection and discrimination deficits

To understand how neural representations of the second target (T2) correlated with behavioral deficits associated with the blink, we constructed distinct dimensions associated with the detection and discrimination of T2. For this, mean EEG traces were computed from 33 bilateral occipito-parietal electrodes (*Figure 6C–D*) to form a multivariate activity representation (vector) at each point in time. These time-varying vectors were subtracted against activity vectors on T2 absent (correct rejection) trials, at each corresponding timepoint and across corresponding inter-target lags, as before, to remove the contribution of the T1-evoked component.

We define the detection dimension ($\boldsymbol{\eta}_{det}$) as the vector mean of each of the T2-present activity vectors measured relative to T2-absent activity (*Figure 6A*); this dimension reflects the change in average EEG representation between T2 present and T2 absent trials. On the other hand, the discrimination dimension ($\boldsymbol{\eta}_{dis}$) represents the vector difference between the two T2-present activity vectors (*Figure 6B*); this dimension reflects the change in EEG representations between the T2 CW and T2 CCW trials. Each of these dimensions was computed separately for each inter-target lag; the magnitude (L2 norm) of each vector ($\| \eta_{det}\|$, $\| \eta_{dis}\|$) represents a scalar quantifying the separability (inverse of the overlap) of the neural representations along the detection or discrimination

dimensions, for the respective lag. Note that these vector magnitudes are equivalently computed as the Euclidean distance between the endpoints of the mean vector representing each class of stimuli ('inter-class distance', *Figure 6C–D*, left). To mitigate biases arising from different numbers of trials across the different lags, we employed bootstrap resampling: for each inter-target lag, we randomly sampled a number of trials corresponding to the lag with the minimum trial count with replacement and computed the mean distances across 10 bootstrap estimates. For quantifying the effect of the attentional blink and for correlation analyses, average inter-class neural distances were computed in a window from 300 to 600ms following T2 onset (*Figure 6C–D*, right). These were then plotted for each lag as a proportion of the inter-class distance for the largest lag (900ms).

## Partial correlations between neural and behavioral metrics

To quantify the neural correlates of blink-induced detection and discrimination deficits, we performed bivariate correlations between neural metrics (ERPs, inter-class distances, and coherences) and the behavioral d' for both tasks. The data were pooled across all lags and participants, and neural and behavioral metrics were normalized for each participant by dividing each measure (neural or behavioral) by the sum of that (respective) measure across all lags. Because the detection d' and discrimination d' in our task were weakly correlated, we performed partial correlations, rather than Pearson's correlations, to identify the specific neural correlate of each; these are reported as '$r_p$' in the Results. To assess the statistical significance of these coefficients, we performed a non-parametric permutation test by shuffling the labels across lags independently for each measure and participant and computing a null distribution over 1000 such random permutations; the p-value reported is the proportion of $r_p$ values in the null distribution that exceeded the partial correlation coefficient observed in the real data. Given our a priori hypothesis about the directionality of correlations, unless stated otherwise, we report one-sided p-values for these correlations; for instance, we expect a positive correlation between P3 ERP amplitude and d' values with increasing inter-target lags. Partial correlations between neural metrics (e.g. between inter-class distance and coherence, 7C-D) were also computed with a similar procedure. Due to multiple comparisons, all p-values derived from permutation tests in partial correlation were adjusted using the Bonferroni-Holm correction test.

## Statistical analyses

For assessing the difference between short-lag and long-lag trials on psychometric or psychophysical metrics (hit, miss, miss identification, false alarm and correct rejections, or d' and c) or neural measures (e.g. ERP amplitudes, coherences, neural distances), non-parametric Wilcoxon signed-rank tests were employed (*signrank* function in Matlab). Unless stated otherwise, p-values in Wilcoxon signed-rank tests were computed using a normal approximation (z-statistic); both the p-value and corresponding z-statistic are reported. All tests were conducted as two-sided tests, unless otherwise specified. We did not perform sample size estimation based on power analysis; our sample size is typical of previous EEG studies of the attention blink (*Sergent et al., 2005*; *Tang et al., 2020*). Nevertheless, to enhance the robustness of our findings, we complemented our statistical analyses by estimating the Bayes factor (BF). BF is particularly valuable in this instance because it quantifies the strength of evidence for the alternative hypothesis over the null hypothesis, rather than relying solely on traditional p-values. The Bayes factor was computed using the JASP toolbox (*JASP team, 2022*); BFs were two-sided unless stated otherwise. Goodness-of-fit of the model was assessed using a randomization test based on the $\chi^2$-statistic; the procedure is described in detail in previous work (*Sridharan et al., 2014*; *Mesulam, 1999*; *Banerjee et al., 2019*; *Sreenivasan and Sridharan, 2019*; *Sagar et al., 2019*). A small p-value (e.g. p<0.05) indicates that the model fit deviated significantly from the observations. Significance testing for the attentional blink effect on sensitivity (d') and criterion (c) parameters was performed with a two-way ANOVA test (*anovan* function in Matlab). We employed robust (bend) correlations (Robust Correlation Toolbox, v2) for conventional correlation analyses, and the *parcorr* function in Matlab for partial correlation analyses. Bonferroni-Holm correction tests were performed with custom code in MATLAB.

## Acknowledgements

This research was supported by a Department of Biotechnology-Wellcome Trust India Alliance Intermediate fellowship, DST Swarna-Jayanti fellowship, a Department of Biotechnology-Indian Institute

of Science Partnership Program grant, and an India-Trento Program for Advanced Research grant (all to DS).

## Additional information

### Funding

| Funder | Grant reference number | Author |
|---|---|---|
| Wellcome Trust DBT India Alliance | IA/I/15/2/502089 | Devarajan Sridharan |
| Department of Science and Technology, Ministry of Science and Technology, India | SB/SJF/2021-22/02 | Devarajan Sridharan |
| An India-Trento Program for Advanced Reserch Grant | INT/ITALY/ITPAR-IV/COG/2018/G | Devarajan Sridharan |
| A Department of Biotechnology-Indian Institute of Science Partnership Program Grant | https://dbtindia.gov.in | Devarajan Sridharan |

The funders had no role in study design, data collection and interpretation, or the decision to submit the work for publication. For the purpose of Open Access, the authors have applied a CC BY public copyright license to any Author Accepted Manuscript version arising from this submission.

### Author contributions

Swagata Halder, Data curation, Formal analysis, Investigation, Visualization, Methodology, Writing – original draft, Writing – review and editing; Deepak Velgapuni Raya, Formal analysis, Visualization, Methodology, Writing – original draft; Devarajan Sridharan, Conceptualization, Resources, Supervision, Funding acquisition, Validation, Visualization, Writing – original draft, Project administration, Writing – review and editing

### Author ORCIDs

Swagata Halder ⓘ http://orcid.org/0000-0002-1807-7116
Deepak Velgapuni Raya ⓘ http://orcid.org/0000-0001-8092-1034
Devarajan Sridharan ⓘ https://orcid.org/0000-0003-1998-9018

### Ethics

All experimental procedures were approved by the Institute Human Ethics Committee, Indian Institute of Science, Bangalore. Participants provided informed, written consent prior to participating in the experiments.

Reviewer #1 (Public review): https://doi.org/10.7554/eLife.97098.3.sa1
Reviewer #2 (Public review): https://doi.org/10.7554/eLife.97098.3.sa2
Reviewer #3 (Public review): https://doi.org/10.7554/eLife.97098.3.sa3
Author response https://doi.org/10.7554/eLife.97098.3.sa4

## Additional files

### Supplementary files

MDAR checklist

Supplementary file 1. Supplementary Information for Distinct neural bases of subcomponents of the attentional blink Supplementary Tables.

## Data availability

Data and custom Matlab code to replicate the experimental findings, as well as the computational modeling results, are available at the following link: https://doi.org/10.17605/OSF.IO/GPZBE.

The following dataset was generated:

| Author(s) | Year | Dataset title | Dataset URL | Database and Identifier |
|-----------|------|---------------|-------------|-------------------------|
| Halder S, Raya DV, Sridharan D | 2025 | Distinct neural bases of subcomponents of the attentional blink | https://doi.org/10.17605/OSF.IO/GPZBE | Open Science Framework, 10.17605/OSF.IO/GPZBE |

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
