## [Editor Report · eLife Assessment]

This study provides an **important** contribution to our understanding of the mechanisms underlying the limited capacity to process rapid sequences of visual stimuli. It reports **convincing** evidence that the attentional blink affects neurally separable processes of visual detection and discrimination. The study will be of interest to neuroscientists and psychologists investigating perception and attention.

---

## [Referee Report · Reviewer #1 (Public review)]

Summary:

In this study, the authors used a multi-alternative decision task and a multidimensional signal-detection model to gain further insight into the cause of perceptual impairments during the attentional blink. The model-based analyses of behavioural and EEG data show that such perceptual failures can be unpacked into distinct deficits in visual detection and discrimination, with visual detection being linked to the amplitude of late ERP components (N2P and P3) and discrimination being linked to coherence of fronto-parietal brain activity.

Strengths:

The strength of this paper lies in the fact that it presents a novel perspective on the cause of perceptual failures during the attentional blink. The multidimensional signal-detection modelling approach is explained clearly, and the results of the study show that this approach offers a powerful method to unpack behavioural and EEG data into distinct processes of detection and discrimination. The discussion of the paper addresses how the findings of separable neural processes involved in detection and discrimination might be linked to extant findings on object recognition and the question of whether the attentional blink involves an all-or-none or gradual impairment in perception.

Weakness:

A minor, unnecessary weakness of the paper is that the authors introduce their study with the aim of determining whether the attentional blink might be due to a criterion shift or to reduced sensitivity in the perceptual process. The criterion shift account remains to be no more than a strawman as the argumentation for this account is weak and easily refuted based on many previous findings. Specifically, the authors suggest that criterion shift might explain the lag-dependent AB effect because participants might be able to infer the lag of a specific trial, thus raising their criterion in case of a short-lag trial, based on factors such as the length of the trial sequence. Importantly, however, attentional blinks have also been observed in many studies in which the sequence length was not indicative of the T1-T2 lag, including - for instance - the many experiments reported in the seminal study by Chun and Potter (1995). The criterion shift account was and remains, therefore, highly implausible and should not have deserved such a prominent role in describing the theoretical motivation for the study.

---

## [Referee Report · Reviewer #2 (Public review)]

Summary:

The authors had two aims: First, to decompose the attentional blink (AB) deficit into the two components of signal detection theory: sensitivity and bias. Second, the authors aimed to assess the two subcomponents of sensitivity: detection and discrimination. They observed that the AB is only expressed in sensitivity. Furthermore, detection and discrimination were doubly dissociated. Detection modulated N2p and P3 ERP amplitude, but not frontoparietal beta-band coherence, whereas this pattern was reversed for discrimination.

Strengths:

The experiment is elegantly designed, and the data -both behavioral and electrophysiological- are aptly analyzed. The outcomes, in particular the dissociation between detection and discrimination blinks, are very consistently and clearly supported by the results. The discussion of the results is also appropriately balanced.

Weaknesses:

The lack of an effect of stimulus contrast does not seem very surprising from what we know of the nature of AB already. Low-level perceptual factors are not thought to cause the AB. This is fine, as there are also other, novel findings reported. In their revision, the authors have bolstered the importance of these (null) findings by referring to AB-specific papers that would have predicted different outcomes in this regard.

The ERP analyses are extended in the revised manuscript, including those of the N1 component, which is now more appropriately analyzed at more lateral electrode sites.

Impact & Context:

The results of this study will likely influence how we think about selective attention in the context of the AB phenomenon. In their revision, the authors have further extended their theoretical framing by referring to recent work on the nature of the AB deficit, showing that it can be discrete (all-or-none) and gradual.

---

## [Referee Report · Reviewer #3 (Public review)]

In the present study, the authors aimed to achieve a better understanding of the mechanisms underlying the attentional blink, that is, a deficit in processing the second of two target stimuli when they appear in rapid succession. Specifically, they used a concurrent detection and identification task in- and outside of the attentional blink and decoupled effects of perceptual sensitivity and response bias using a novel signal detection model. They conclude that the attentional blink selectively impairs perceptual sensitivity but not response bias, and link established EEG markers of the attentional blink to deficits in stimulus detection (N2p, P3) and discrimination (fronto-parietal high-beta coherence), respectively. Taken together, their study suggests distinct mechanisms mediating detection and discrimination deficits in the attentional blink.

This innovative study appears to have been carefully conducted and the overall conclusions seem warranted given the results. In my opinion, the manuscript is a valuable contribution to the current literature on the attentional blink. Moreover, the novel paradigm and signal detection model are likely to stimulate future research.

Major strengths of the present study include its innovative approach to investigating the mechanisms underlying the attentional blink, an elegant, carefully calibrated experimental paradigm, a novel signal detection model, multifaceted data analyses using state-of-the-art model comparisons and robust statistical tests, and an interesting discussion on the neural mechanisms underlying detection versus identification.

Weaknesses concern a lack of clarity regarding specific statistical hypotheses and correction for multiple comparisons (e.g., across or within the multiple classes of tests) in the Methods, relatively low statistical power (N = 24/18 for behavioral/ERP data, respectively), unusual and heavy EEG filtering (0.5-18 Hz bandpass and 9-11 Hz bandstop), data-driven analyses (e.g., pooling of lag 1 and 3 trials a posteriori), and the absence of a discussion of limitations.

---

## [Author Response]

The following is the authors’ response to the original reviews

**Reviewer #1 (Public review):**
Summary:In this study, the authors used a multi-alternative decision task and a multidimensional signaldetection model to gain further insight into the cause of perceptual impairments during the attentional blink. The model-based analyses of behavioural and EEG data show that such perceptual failures can be unpacked into distinct deficits in visual detection and discrimination, with visual detection being linked to the amplitude of late ERP components (N2P and P3) and discrimination being linked to the coherence of fronto-parietal brain activity.Strengths:The main strength of this paper lies in the fact that it presents a novel perspective on the cause of perceptual failures during the attentional blink. The multidimensional signal detection modelling approach is explained clearly, and the results of the study show that this approach offers a powerful method to unpack behavioural and EEG data into distinct processes of detection and discrimination.

Thank you.

Weaknesses:(1.1) While the model-based analyses are compelling, the paper also features some analyses that seem misguided, or, at least, insufficiently motivated and explained. Specifically, in the introduction, the authors raise the suggestion that the attentional blink could be due to a reduction in sensitivity or a response bias. The suggestion that a response bias could play a role seems misguided, as any response bias would be expected to be constant across lags, while the attentional blink effect is only observed at short lags. Thus, it is difficult to understand why the authors would think that a response bias could explain the attentional blink.

In the revision, we seek to better motivate the bias component. A deficit in T2 identification accuracy could arise from either sensitivity or criterion effects at short lags. For example, in short T1-T2 lag trials participants may adopt a more conservative choice criterion for reporting the presence of T2 thereby yielding lower accuracies for short lags. Criterion effects need not be uniform across lags: A participant could infer the T1-T2 lag on each trial based on various factors, such as trial length, and systematically adjust their choice criterion across lags, prior to making a response.

Below, we present a simple schematic for how a conservative choice criterion impacts accuracy. Consider a conventional attentional blink paradigm where the task is to detect and report T2's presence. For simplicity, we assume that prior probabilities for T2’s occurrence are equal, such that the number of “T2 present” and “T2 absent” trials are equal.

We model this task with a one-dimensional signal detection theory (SDT) model (left panel). Here, ψ represents the decision variable and the red and gray Gaussians represent the conditional density of ψ for the T2 present (“signal”) and T2 absent (“noise”) conditions, respectively. We increase the criterion from its optimal value (here, midpoint of signal and noise means), to reflect increasingly conservative choices. As the criterion increases and deviates further from its optimal value – here, reflecting a conservative bias – accuracy drops systematically (right panel).

**Author response image 1. sa4fig1:** 

We have revised the Introduction as follows:

“Distinguishing between sensitivity and criterion effects is crucial because a change in either of these parameters can produce a change in the proportion of correct responses[41,42]. A lower proportion of correct T2 detections may reflect not only a lower detection d’ at short lags but also a sub-optimal choice criterion corresponding, for instance, to a conservative detection bias (Fig. 1, right, top). Importantly, such criterion effects need not be uniform across intertarget lags: the lag on each trial could be inferred based on various factors, such as trial length, allowing participants to adopt different choice criteria for the different lags prior to making a response.”

(1.2) A second point of concern regards the way in which the measures for detection and discrimination accuracy were computed. If I understand the paper correctly, a correct detection was defined as either correctly identifying T2 (i.e., reporting CW or CCW if T2 was CW or CCW, respectively, see Figure 2B), or correctly reporting T2's absence (a correct rejection).Here, it seems that one should also count a misidentification (i.e., incorrect choice of CW or CCW when T2 was present) as a correct detection, because participants apparently did detect T2, but failed to judge/remember its orientation properly in case of a misidentification. Conversely, the manner in which discrimination performance is computed also raises questions. Here, the authors appear to compute accuracy as the average proportion of T2present trials on which participants selected the correct response option for T2, thus including trials in which participants missed T2 entirely. Thus, a failure to detect T2 is now counted as a failure to discriminate T2. Wouldn't a more proper measure of discrimination accuracy be to compute the proportion of correct discriminations for trials in which participants detected T2?

Indeed, detection and discrimination accuracies were computed with precisely the same procedure, and under the same conditions, as described by the Reviewer. We regret our poor description. For clarity, we have revised the following line in the Results section; we have also updated the Methods (section on Behavioral data analysis: Measuring attentional blink effects on psychometric quantities).

“Detection accuracies were calculated based on the proportion of trials in which T2 was correctly detected (Methods). Briefly, we computed the average proportion of hits, misidentifications, and correct rejections; misidentifications were included because, although incorrectly identified, the target was nevertheless correctly detected. In contrast, discrimination accuracies were derived from T2 present trials, based on the proportion of correct identifications alone (Methods).”

(1.3) My last point of critique is that the paper offers little if any guidance on how the inferred distinction between detection and discrimination can be linked to existing theories of the attentional blink. The discussion mostly focuses on comparisons to previous EEG studies, but it would be interesting to know how the authors connect their findings to extant, mechanistic accounts of the attentional blink. A key question here is whether the finding of dissociable processes of detection and discrimination would also hold with more meaningful stimuli in an identification task (e.g., the canonical AB task of identifying two letters shown amongst digits).There is evidence to suggest that meaningful stimuli are categorized just as quickly as they are detected (Grill-Spector & Kanwisher, 2005; Grill-Spector K, Kanwisher N. Visual recognition: as soon as you know it is there, you know what it is. Psychol Sci. 2005 Feb;16(2):152-60. doi: 10.1111/j.0956-7976.2005.00796.x. PMID: 15686582.). Does that mean that the observed distinction between detection and discrimination would only apply to tasks in which the targets consist of otherwise meaningless visual elements, such as lines of different orientations?

Our results are consistent with previous literature suggested by the reviewer. Specifically, we model detection and discrimination not as sequential processes, but as concurrent computations (Figs. 3A-B). Yet, our results suggest that these processes possess distinct neural bases. We have further revised the Discussion in context of this literature in the revised manuscript.

“…Interestingly, we found no evidence indicating that these two computations (detection and discrimination) were sequential; in fact, the modulation of beta coherence occurred almost immediately after T2 onset, and lasted well afterwards (>400 ms from T2 onset) (Fig. 5A-B) suggesting that an analysis of T2’s features proceeded in parallel with its detection and consolidation. We also modeled detection and discrimination as concurrent computations in our SDT model (Fig. 3A-B). Previous work suggests that while object detection and categorization processes proceed in parallel, detection and identification processes occur sequentially[77]. Our results are in line with this literature, if we consider T2’s discrimination judgement – clockwise versus counterclockwise of vertical – to be a categorization, rather than an identification judgement. Moreover, this earlier study[75] observed significant trial-wise correlations between detection and categorization responses, suggesting that the two processes involve the operation of the same perceptual filters (“analyzers”). Our study, on the other hand, reports distinct neural bases for detection and discrimination computations. Yet, the two sets of findings are not mutually contradictory.

In many conventional attentional blink tasks[3,20,25], complex visual stimuli, like letters, must be detected among a stream of background distractors with closely similar features, such as digits. In this case, target detection would require the operation of shape-selective perceptual filters for feature analysis. These same shape-selective filters would be involved also for discriminating between distinct, but related target stimuli (e.g., two designated candidate letters). In our task, target gratings needed to be distinguished in a stream of plainly distinct background distractors (plaids), whereas the discrimination judgement involved analysis of grating orientation. As a result, our task design likely precludes the need for the same perceptual filters in the detection and the discrimination judgements. Absent this common feature analysis, our results suggest distinct electrophysiological correlates for the detection and discrimination of targets.”

**Reviewer #2 Public review:**
Summary:The authors had two aims: First, to decompose the attentional blink (AB) deficit into the two components of signal detection theory; sensitivity and bias. Second, the authors aimed to assess the two subcomponents of sensitivity; detection and discrimination. They observed that the AB is only expressed in sensitivity. Furthermore, detection and discrimination were doubly dissociated. Detection modulated N2p and P3 ERP amplitude, but not frontoparietal beta-band coherence, whereas this pattern was reversed for discrimination.Strengths:The experiment is elegantly designed, and the data - both behavioral and electrophysiological - are aptly analyzed. The outcomes, in particular the dissociation between detection and discrimination blinks, are consistently and clearly supported by the results. The discussion of the results is also appropriately balanced.

Thank you.

Weaknesses:(2.1) The lack of an effect of stimulus contrast does not seem very surprising from what we know of the nature of AB already. Low-level perceptual factors are not thought to cause AB. This is fine, as there are also other, novel findings reported, but perhaps the authors could bolster the importance of these (null) findings by referring to AB-specific papers, if there are indeed any, that would have predicted different outcomes in this regard.

While there is consensus that the low-level perceptual factors are not affected by the attentional blink, other studies have suggested evidence to the contrary (e.g., Chua et al, Percept. Psychophys., 2005)[1]. We have mentioned the significance of our findings in the context of such conflicting evidence in literature, in the revised Discussion.

“Surprisingly, we found no significant effect of contrast on either type of deficit (Figs. 2A-B). In other words, high (100%) contrast T2 stimuli were also strongly susceptible to the detection and discrimination bottlenecks associated with the attentional blink. Thus, despite a clear contrast-dependent encoding of T2 in early sensory cortex, the attentional blink produced a significant deficit with downstream processing, even for targets of high contrast. While at odds with some earlier work, which suggest an early-stage perceptual bottleneck [82–84], these results are largely consistent with findings from the majority of previous studies [3,7,9,11,19,20,82,85,86] which suggest a late-stage bottleneck.”

(2.2) On an analytical note, the ERP analysis could be finetuned a little more. The task design does not allow measurement of the N2pc or N400 components, which are also relevant to the AB, but the N1 component could additionally be analyzed. In doing so, I would furthermore recommend selecting more lateral electrode sites for both the N1, as well as the P1. Both P1 and N1 are likely not maximal near the midline, where the authors currently focused their P1 analysis.

We performed these suggested analysis. Whereas in the original submission we had used the O1, O2 and Oz electrodes, we now estimate the P1 and N1 with the more lateral P7 and P8 electrodes[2], as suggested by the reviewer.

Even with these more lateral electrodes, we did not observe a significant N1 component in a 90-160 ms window[3] in the long lag trials (p=0.207, signed rank test for amplitude less than zero); a one-tailed Bayes factor (BF=1.35) revealed no clear evidence for or against an N1 component. Analysis of the P1 component with these more lateral electrodes also yielded no statistically significant blink-induced modulation (P1(short lag-long lag) = 0.25 ± 0.16, uV, p=0.231, BF=0.651) (SI Figure S3, revised).

These updated analyses are now reported in the revised Results (lines 317-319) and Methods (lines 854-855). In addition, we have revised SI Table S2 with the new P1 component analysis.

(2.3) Impact & Context:The results of this study will likely influence how we think about selective attention in the context of the AB phenomenon. However, I think its impact could be further improved by extending its theoretical framing. In particular, there has been some recent work on the nature of the AB deficit, showing that it can be discrete (all-or-none) and gradual (Sy et al., 2021; Karabay et al., 2022, both in JEP: General). These different faces of target awareness in the AB may be linked directly to the detection and discrimination subcomponents that are analyzed in the present paper. I would encourage the authors to discuss this potential link and comment on the bearing of the present work on these behavioural findings.

Thank you. We have now discussed our findings in the context of these recent studies in the revised manuscript.

“…In line with this hypothesis, we discovered that the attentional blink induced dissociable detection and discrimination deficits. There was no statistically significant correlation between these two types of deficits within and across participants and evidence for such a correlation was weak, at best. Unlike previous target identification designs that conflated attentional blink’s effect on detection versus discrimination performance[3,4,9,25,37], our 3-AFC task, and associated signal detection model enabled quantifying each of these deficits separately and identifying a double dissociation between their respective neural correlates. Our dissociation of the attentional blink into distinct subcomponents is complementary to recent studies, which examined whether the attentional blink reflects an all-or-none phenomenon[73,74]. For example, the T2 deficit induced by the attentional blink can be either all-or-none or graded, depending on whether T1 and T2 judgements involve distinct or common features, respectively[73]. While a graded change in precision could reflect sensitivity effects, an all-or-none change in guess rates – without a concomitant change in precision – may reflect a criterion increase (conservative detection bias) effect. Future experiments that incorporate a three-alternative response, with concurrent detection and discrimination, along with key task elements of these earlier studies, may further help resolve these findings.”

**Reviewer #3 (Public review):**
Summary:In the present study, the authors aimed to achieve a better understanding of the mechanisms underlying the attentional blink, that is, a deficit in processing the second of two target stimuli when they appear in rapid succession. Specifically, they used a concurrent detection and identification task in- and outside of the attentional blink and decoupled effects of perceptual sensitivity and response bias using a novel signal detection model. They conclude that the attentional blink selectively impairs perceptual sensitivity but not response bias, and link established EEG markers of the attentional blink to deficits in stimulus detection (N2p, P3) and discrimination (fronto-parietal high-beta coherence), respectively. Taken together, their study suggests distinct mechanisms mediating detection and discrimination deficits in the attentional blink.Strengths:Major strengths of the present study include its innovative approach to investigating the mechanisms underlying the attentional blink, an elegant, carefully calibrated experimental paradigm, a novel signal detection model, and multifaceted data analyses using state-of-the art model comparisons and robust statistical tests. The study appears to have been carefully conducted and the overall conclusions seem warranted given the results. In my opinion, the manuscript is a valuable contribution to the current literature on the attentional blink. Moreover, the novel paradigm and signal detection model are likely to stimulate future research.

Thank you.

Weaknesses:Weaknesses of the present manuscript mainly concern the negligence of some relevant literature, unclear hypotheses, potentially data-driven analyses, relatively low statistical power, potential flaws in the EEG methods, and the absence of a discussion of limitations. In the following, I will list some major and minor concerns in detail.(3.1) Hypotheses: I appreciate the multifaceted, in-depth analysis of the given dataset including its high amount of different statistical tests. However, neither the Introduction nor the Methods contain specific statistical hypotheses. Moreover, many of the tests (e.g., correlations) rely on selected results of previous tests. It is unclear how many of the tests were planned a priori, how many more were performed, and how exactly corrections for multiple tests were implemented. Thus, I find it difficult to assess the robustness of the results.

We hypothesized that neural computations associated with target detection would be characterized by regional (local) neuronal markers (e.g., parietal or occipital ERPs), whereas computations linked to feature discrimination would involve neural coordination across multiple brain regions (e.g. fronto-parietal coherence) (lines 135-138). We planned and conducted our statistical tests based on this hypothesis. All multiple comparison corrections (Bonferroni-Holm correction, see Methods) were performed separately for each class of analyses.

Based on this overarching hypothesis, the following tests were planned and conducted.

ERP analysis: Based on an extensive review of recent literature[4] Zivony et al., 2022 we performed the following tests: (i) We tested whether four ERP component amplitudes (parietal P1, fronto-central P2, occipito-parietal N2p, and parietal P3) were significantly different between short and long lags with a Wilcoxon signed-rank test followed by Bonferroni-Holm multiple comparison correction; (ii) We correlated the ERPs whose amplitudes showed a significant difference in analysis (i) with detection and discrimination d’ deficits (six correlations) using robust (bend) correlations[5]; again, this was followed by a Bonferroni-Holm multiple comparison correction. Note that there is no circularity with planning analysis (ii) based on the results of analysis (i) because the latter is agnostic to detection versus discrimination blink deficits. In case (i), where no a priori hypothesis about directionality were available, all p-values were based on two-tailed tests but for case (ii), where we had an a priori directional hypothesis, p-values were computed from one-tailed tests. This has now been clarified in the revised Methods lines 937-940 and 950-952.

Coherence analysis: Based on a seminal study of long-range synchrony modulation by the attentional blink[6], we examined fronto-parietal coherence in the beta (13-30 Hz) band, separately for the left and right hemispheres, and performed the following comparisons. (i) We computed differences between the fronto-parietal coherogram (time-frequency representation of coherence, Fig. 5A-D) between short-lag and long-lag conditions, and performed a twodimensional cluster-based permutation test[7]; this method inherently corrects for multiple comparisons across time-frequency windows. (ii) Because the analysis in (i) revealed the clearest evidence for coherence differences in the canonical high-beta (20-30 Hz band) in the left fronto-parietal electrodes (Figs. 5C-D; 0-300 ms following target onset), we correlated power in this band with detection and discrimination d’ deficits; this was followed by a Bonferroni-Holm multiple comparison correction. As before there is no circularity with planning analysis (ii) based on the results of analysis (i) because the latter is agnostic to detection versus discrimination blink deficits. Again, in case (i), where no a priori hypothesis about directionality was made, all p-values were based on two-tailed tests but for case (ii), where we had an a priori directional hypothesis, p-values were computed from one-tailed tests.

For completeness, we performed all of the other correlations, for example, correlations with coherence in the low-beta band or with the right fronto-parietal electrodes (SI Table 3). These latter analyses were not planned, nor did they yield significant results.

Neural distance analysis: This was a novel analysis designed to test the hypothesis that detection and discrimination deficits would be correlated with neural distances along distinct dimensions. (i) First, we compared neural distances across lag conditions at different timepoints following target onset with a one-dimensional cluster-based permutation test[7] ; (ii) Next, we correlated the neural distances along the detection and discrimination dimension with the detection and discrimination d’ deficits (Fig. 6E-F, 6G-H), as well as with the ERP and coherence markers (Fig. 7A-B, 7C-D). For each of these analyses, we employed robust (bend) correlations[5] followed by a Bonferroni-Holm multiple comparison correction. As before, pvalues were computed using two-tailed tests for case (i) and one-tailed tests for case (ii), based on the absence or presence of an a priori directional hypothesis.

(3.2) Power: Some important null findings may result from the rather small sample sizes of N = 24 for behavioral and N = 18 for ERP analyses. For example, the correlation between detection and discrimination d' deficits across participants (r=0.39, p=0.059) (p. 12, l. 263) and the attentional blink effect on the P1 component (p=0.050, no test statistic) (p. 14, 301) could each have been significant with one more participant. In my opinion, such results should not be interpreted as evidence for the absence of effects.

We have modified these claims in the revised Results. In addition, we now compute and report Bayes factors, which enable evaluating evidence for the presence versus absence of effects.

“Detection and discrimination d’ deficits were not statistically significantly correlated (r=0.39, t=2.28, p=0.059); Bayes factor analysis revealed no clear evidence for or against a correlation between these subcomponent deficits (BF=1.18) (SI Fig. S2, left).”

“Discrimination accuracy deficits were not statistically significantly different between high and low detection accuracy deficit blocks (z=1.97, p=0.067), and the Bayes factor revealed no strong evidence for or against such a difference (BF=1.42) (Fig. 3G).”

In addition, the results are interpreted as follows (lines 294-296):

“Moreover, detection and discrimination d’ deficits were not significantly correlated both within and across participants, with no clear evidence for or against a correlation, based on the Bayes factor.”

The null result on the P1 has changed because of the analysis with the alternative electrode set suggested by Reviewer #2 (see comment #2.2). We now report these results as follows:

“By contrast, the P1, an early sensory component, showed no statistically significant blinkinduced modulation (P1 = 0.25 ± 0.16µV, z = 1.19, p=0.231, BF = 0.651) (SI Fig. S3).”

(3.3) Neural basis of the attentional blink: The introduction (e.g., p. 4, l. 56-76) and discussion (e.g., p. 19, 427-447) do not incorporate the insights from the highly relevant recent review by Zivony & Lamy (2022), which is only cited once (p. 19, l. 428). Moreover, the sections do not mention some relevant ERP studies of the attentional blink (e.g., Batterink et al., 2012; Craston et al., 2009; Dell'Acqua et al., 2015; Dellert et al., 2022; Eiserbeck et al., 2022; Meijs et al., 2018).

We have now cited these previous studies at the appropriate places in the revised Introduction.

“The effect of the attentional blink on the processing of the second target is well studied. In particular, previous studies have investigated the stage at which attentional blink affects T2’s processing (early or late) [14–17] and the neural basis of this effect, including the specific brain regions involved[15,18–20]. Several theoretical frameworks characterize a sequence of phases of the attentional blink, including target selection based on relevance, detection, feature processing, and encoding into working memory[9,21]. Overall, there is little support for attentional blink deficits at an early, sensory encoding[14] stage; by contrast, the vast majority of literature suggests that T2’s processing is affected at a late stage[8,10]. Consistent with these behavioral results, scalp electroencephalography (EEG) studies have reported partial or complete suppression of late event-related potential (ERP) components, particularly those linked to attentional engagement (P2, N2, N2pc or VAN)[15,22–25], working memory (P3) [20,26–30] or semantic processing (N400)[31]; early sensory components (P1/N1) are virtually unaffected[20,24] (reviewed in detail in Zivony and Lamy, 2022[32]) .”

(3.4) Detection versus discrimination: Concerning the neural basis of detection versus discrimination (e.g., p. 6, l. 98-110; p. 18, l. 399-412), relevant existing literature (e.g., Broadbent & Broadbent, 1987; Hillis & Brainard, 2007; Koivisto et al., 2017; Straube & Fahle, 2011; Wiens et al., 2023) is not included.

Thank you for these suggestions. We have now cited these studies in the revised Discussion.

“It is increasingly clear that detection and discrimination are separable processes, each mediated by distinct neural mechanisms. Behaviorally, accurately identifying the first target, versus merely detecting it, produces stronger deficits with identifying the second target[59]. Moreover, dissociable mechanisms have been reported to mediate object detection and discrimination in visual adaptation contexts[60]. Neurally, shape detection and identification judgements produce activations in non-overlapping clusters in various brain regions in the visual cortex, inferior parietal cortex, and the medial frontal lobe[61]. Similarly, occipital ERPs associated with conscious awareness also show clear differences between detection and discrimination. For instance, an early posterior negative component (200-300 ms) was significantly modulated in amplitude by success in detection, but not in identification[62]. The closely related visual awareness negativity (VAN) was substantially stronger at the detection, compared to the discrimination, threshold[63].

Furthermore, a significant body of previous work has reported dissociable behavioural and neural mechanisms underlying attention’s effects on target detection versus discrimination. Behavioral studies have reported distinct effects on target detection versus discrimination in both endogenous[64] and exogenous[65] attention tasks.”

(3.5) Pooling of lags and lags 1 sparing: I wonder why the authors chose to include 5 different lags when they later pooled early (100, 300 ms) and late (700, 900 ms) lags, and whether this pooling is justified. This is important because T2 at lag 1 (100 ms) is typically "spared" (high accuracy) while T2 at lag 3 (300 ms) shows the maximum AB (for reviews, see, e.g., Dux & Marois, 2009; Martens & Wyble, 2010). Interestingly, this sparing was not observed here (p. 43, Figure 2). Nevertheless, considering the literature and the research questions at hand, it is questionable whether lag 1 and 3 should be pooled.

Lag-1 sparing is not always observed in attentional blink studies; there are notable exceptions to reports of lag-1 sparing[8,9]. Our statistical tests revealed no significant difference in accuracies between short lag (100 and 300 ms) trials or between long lag (700 and 900 ms) trials but did reveal significant differences between the short and long lag trials (ANOVA, followed by post-hoc tests). To simplify the presentation of the findings, we pooled together the short lag (100 and 300 ms) and, separately, the long lag (700 and 900 ms) trials. We have presented these analyses, and clarified the motivation for pooling these lags in the revised Methods.

“Based on these psychometric measures, we computed detection and discrimination accuracies as follows. Detection accuracies were computed as the average proportion of the hits, misidentification and correct rejection responses; misidentifications were included because not missing the target reflected accurate detection. By contrast, discrimination accuracies were computed based on the average proportion of the two correct identifications (hits) on T2 present trials alone. We performed 2-way ANOVAs on both detection and discrimination accuracies with the inter-target lag (5 values) and T2 contrast independent factors. We found main effects of both lag (F(4,92)=18.81, p<0.001) and contrast (F(1,92)=21.78, p<0.001) on detection accuracy, but no interaction effect between lag and contrast (F(4,92)=1.92, p=0.113). Similarly, we found main effects of both lag (F(4,92)=25.08, p<0.001) and contrast (F(1,92)=16.58, p<0.001) on discrimination accuracy, but no interaction effect between lag and contrast (F(4,92)=0.93, p=0.450). Post-hoc tests based on Tukey’s HSD revealed a significant difference in discrimination accuracies between the two shortest lags (100 ms and 300 ms) and the two longest lags (700 and 900 ms) for both low and high contrast targets, and for both detection and discrimination accuracies (p<0.01). But they revealed no significant difference between the two shortest lags (p>0.25) or the two longest lags (p>0.40) for either target contrast or for either accuracy type. As a result, for subsequent analyses, we pooled together the “short lag” (100 ms and 300 ms) and the “long lag” (700 ms and 900 ms) trials. We quantified the effect of the attentional blink on each of the psychometric measures as well as detection and discrimination accuracies by comparing their respective, average values between the short lag and long lag trials, separately for the high and low T2 contrasts.”

(3.6) Discrimination in the attentional blink. Concerning the claims that previous attentional blink studies conflated detection and discrimination (p. 6, l. 111-114; p. 18, l. 416), there is a recent ERP study (Dellert et al., 2022) in which participants did not perform a discrimination task for the T2 stimuli. Moreover, since the relevance of all stimuli except T1 was uncertain in this study, irrelevant distractors could not be filtered out (cf. p. 19, l. 437). Under these conditions, the attentional blink was still associated with reduced negativities in the N2 range (cf. p. 19, l. 427-437) but not with a reduced P3 (cf. p. 19, l 439-447).

We have addressed the relationship between our findings and those of Dellert et al (2022)[10] in the revised Discussion.

“… In the present study, we observed that the parietal P3 amplitude was correlated selectively with detection, rather than discrimination deficits. This suggests that the P3 deficit indexes a specific bottleneck with encoding and consolidating T2 into working memory, rather than an inability to reliably maintain its features. In this regard, a recent study[22] measured ERP correlates of the perceptual awareness of the T2 stimulus whose relevance was uncertain at the time of its presentation. In contrast to earlier work, this study observed no change in P3b amplitude across seen (detected) and unseen targets. Taken together with this study, our findings suggest that rather than indexing visual awareness, the P3 may index detection, but only when information about the second target, or a decision about its appearance, needs to be maintained in working memory. Additional experiments, involving targets of uncertain relevance, along with our behavioral analysis framework, may help further evaluate this hypothesis.”

(3.7) General EEG methods: While most of the description of the EEG preprocessing and analysis (p. 31/32) is appropriate, it also lacks some important information (see, e.g., Keil et al., 2014). For example, it does not include the length of the segments, the type and proportion of artifacts rejected, the number of trials used for averaging in each condition, specific hypotheses, and the test statistics (in addition to p-values).

We regret the lack of details. We have included these in the revised Methods, and expanded on the description of the trial rejection (SCADS) algorithm.

The revised Methods section on EEG Preprocessing mentions the type and proportion of artifacts rejected:

“We then epoched the data into trials and applied SCADS (Statistical Control of Artifacts in Dense Array EEG/MEG Studies[90]) to identify bad epochs and artifact contaminated channels. SCADS detects artifacts based on three measures: maximum amplitude over time, standard deviation over time, and first derivative (gradient) over time. Any electrode or trial exhibiting values outside the specified boundaries for these measures was excluded. The boundaries were defined as M ± n*λ, where M is the grand median across electrodes and trials for each of the three measures, and λ is the root mean square (RMS) of the deviation of medians across sensors relative to the grand median. We set n to 3, allowing data within three boundaries to be retained. The percentage of electrodes per participant rejected was 6.3 ± 0.43% (mean ± s.e.m. across participants), whereas the percentage of trials rejected per electrode and participant was 3.4 ± 0.33% (mean ± s.e.m.).”

The revised Methods section on ERP analysis mentions the number of trials for averaging in each condition and the length of the segments:

“First trials were sorted based on inter-target lags (100, 300, 500, 700 and 900 ms). This yielded an average of (200±13, 171±9.71, 145 ± 7.54, 117 ± 5.43, 87 ± 4.51) (mean ± s.e.m. across participants) trials for each of the 5 lags, respectively.”

“Then, EEG traces were epoched from -300 ms before to +700 ms after either T1 onset or T2 onset and averaged across trials to estimate T1-evoked and T2-evoked ERPs, respectively.”

Specific hypotheses are mentioned in response #3.1; we also now mention the test statistic associated with each test at the appropriate places in the Results. For example:

“Among these ERP components, the N2p component and the P2 component were both significantly suppressed during the blink (∆amplitude, short-lag – long-lag: N2p=-0.47 ± 0.12 µV, z=-3.20, p=0.003, BF=40, P2=-0.19 ± 0.07 µV, z=-2.54, p=0.021, BF=4.83, signed rank test) (Fig. 4A, right). Similarly, the parietal P3 also showed a significant blink-induced suppression (P3 = -0.45 ± 0.09µV, z=-3.59, p < 0.001, BF>10^2^) (Fig. 4B, right).”

“Neural inter-class distances (||η||) along both the detection and discrimination dimensions decreased significantly during the blink (short lag-long lag: ∆||ηdet|| = -1.30 ± 0.70, z=-3.68, p=0.006, BF=20; ∆||ηdis|| = -1.23 ± 0.42, z=-3.54, p<0.001, BF>10^2^) (Figs. 6C-D).”

(3.8) EEG filters: P. 31, l. 728: "The data were (...) bandpass filtered between 0.5 to 18 Hz (...). Next, a bandstop filter from 9-11 Hz was applied to remove the 10 Hz oscillations evoked by the RSVP presentation." These filter settings do not follow common recommendations and could potentially induce filter distortions (e.g., Luck, 2014; Zhang et al., 2024). For example, the 0.5 high-pass filter could distort the slow P3 wave. Mostly, I am concerned about the bandstop filter. Since the authors commendably corrected for RSVP-evoked responses by subtracting T2-absent from T2-present ERPs (p. 31, l. 746), I wonder why the additional filter was necessary, and whether it might have removed relevant peaks in the ERPs of interest.

Thank you for this suggestion. Originally, the 9-11 Hz bandstop filter was added to remove the strong 10 Hz evoked oscillation from the EEG response for obtaining a cleaner signal for the other analyses, like the analysis of neural dimensions (Fig. 6)

We performed two control ERP analyses to address the reviewers’ concern:

(1) We removed the bandstop filter and re-evaluated the P1, P2, N2pc and P3 ERP amplitudes. We observed no statistically significant difference in the modulation of any of the 4 ERP components (P1: p=0.031, BF=0.692, P2: p=0.038, BF=1.21, N2pc: p=0.286, BF=0.269, P3: p=0.085, BF=0.277). In particular, Bayes Factor analysis revealed substantial evidence against a difference in the N2pc and P3 amplitudes before versus after the bandstop filter removal (BF<0.3).

(2) We removed the bandstop filter and repeated all of the same analyses as reported in the Results and summarized in SI Table S2. We observed a virtually identical pattern of results, summarized in an analogous table, below (compare with SI Table S2, revised, in the Supplementary Information).

**Author response table 1. sa4table1:** 

ERP components	Blink deficit (mean +- s.e.m)	Partial correlation with sensitivity (d')	
		Partial correlation with detection d'	Partial correlation with discrimination d'
P1 (p7-p8)	-0.12+-0.21, p = 0.616	r_(p)=0.16 p=0.325	r_(p)=-0.11 p=0.390
P2 (frontocentral)	0.26+-0.09, p = 0.018	r_(p)=0.00p=0.742	r_(p)=0.25p=0.264
N2p (occipitoparietal)	-0.51+-0.14p=0.018	r_(p)=-0.31p < 0.001	r_(p)=-0.05p=0.742
P3 (parietal)	0.50+-0.13,p=0.008	r_(p)=0.32p < 0.001	r_(p)=0.15p=0.388

We have now mentioned this control analysis briefly in the Methods (lines 863-865).

(3.9) Coherence analysis: P. 33, l. 786: "For subsequent, partial correlation analyses of coherence with behavioral metrics and neural distances (...), we focused on a 300 ms time period (0-300 ms following T2 onset) and high-beta frequency band (20-30 Hz) identified by the cluster-based permutation test (Fig. 5A-C)." I wonder whether there were any a priori criteria for the definition and selection of such successive analyses. Given the many factors (frequency bands, hemispheres) in the analyses and the particular shape of the cluster (p. 49, Fig 5C), this focus seems largely data-driven. It remains unclear how many such tests were performed and whether the results (e.g., the resulting weak correlation of r = 0.22 in one frequency band and one hemisphere in one part of a complexly shaped cluster; p. 15, l. 327) can be considered robust.

Please see responses to comments #3.1 and #3.2 (above). In addition to reporting further details regarding statistical tests, their hypotheses, and multiple comparisons corrections, we computed Bayes factors to quantify the strength of the evidence for correlations, as appropriate. Interpretations have been rephrased depending on whether the evidence for the null or alternative hypothesis is strong or equivocal. For example:

“Bayes factor analysis revealed no clear evidence for or against a correlation between these subcomponent deficits (BF=1.18) (SI Fig. S2, left).”

“Discrimination accuracy deficits were not statistically significantly different between high and low detection accuracy deficit blocks (z=1.97, p=0.067), and the Bayes factor revealed no strong evidence for or against such a difference (BF=1.42) (Fig. 3G).”

**Recommendations for the authors:**

**Reviewer #1 (Recommendations for the authors):**
(1.a) Line 76-79: "Despite this extensive literature, previous studies have essentially treated the attentional blink as a unitary, monolithic phenomenon. As a result, fundamental questions regarding the component mechanisms of the attentional blink remain unanswered." This statement seems antithetical to the fact that theories of the AB suggest a variety of different mechanisms as possible causes of the effect.

The statement has been revised as follows:

“Despite this extensive literature, many previous studies have[studied the attentional blink as a unitary phenomenon. While some theoretical models9,21,32] and experimental studies[38,39] have explored distinct mechanisms underlying the attentional blink, several fundamental questions about its distinct component mechanisms remain unanswered.”

(1.b) Line 95-97: Here, the authors should explain in more detail how a response bias could fluctuate across lags.

Addressed in response to public reviews, #1.1.

(1.c) Line 98: I found this second question a much more compelling motivation for the study than the earlier stated question of whether the AB reflects a reduction in sensitivity or a fluctuation (?) of response bias.

Thank you.

(1.d) Line 143: What do the authors mean by "geometric" distribution of lags? In virtually all AB studies, the distribution of lags is uniform. Wasn't that the case in this study?

We employed a geometric distribution for the trials of different lags, and verified that the sampled distribution of lags was well fit by this distribution (χ^2^(3, 312)=0.22, p=0.974). We chose a geometric distribution – with a flat hazard function[11] – over the uniform distribution to avoid conflating the effects of temporal expectation with those of the attention blink on criterion[12] at different lags.

(1.e) Line 158-160: Explain why incorrect discrimination responses were not counted as correct detection. Explain why failure to detect T2 was counted as a discrimination error.

Addressed in response to public reviews, #1.2.

(1.f) Line 167: The results do not show lag-1 sparing, which is a typical property of the AB.The authors should report this, and explain why their paradigm did not show a sparing effect.

Addressed in response to public reviews, #3.5.

(1.g) Line 262-263: With only 24 participants, the study appears to be underpowered to reliably detect correlations. This should be noted as a limitation.

Addressed in response to public reviews, #3.2.

(1.h) Line 399-412: This section could be moved to the introduction to explain and motivate the aim of examining the distinct contributions of detection and discrimination to the AB.

We have revised the Introduction to better motivate the aims of the study.

**Reviewer #2 (Recommendations for the authors):**
(2.a) A small note about the writing: as a matter of style, I would advise editing the generic phrasing (e.g., "shedding new light", "complex interplay") in abstract and general discussion.

These are now revised as follows (for example):

Line 26 - “These findings provide detailed insights into the subcomponents of the attentional blink….”

Line 596 - “More broadly, these findings contribute to our understanding of the relationship between attention and perception….”

(2.b) Some references appear double and/or without volume or page numbers (e.g., 44/61).

Thank you. Amended now.

**Reviewer #3 (Recommendations for the authors):**
(3.a) Suggestions for additional analyses:I appreciate that the authors have quantified the evidence for null effects in simple comparisons using Bayes factors. In my opinion, the study would additionally benefit from Bayesian ANOVAs, which can also easily be implemented in JASP (Keysers et al., 2020), which the authors have already used for the other tests. As a result, they could further substantiate some of their claims related to null effects (e.g., p. 9, l. 175; p. 12, l. 246).

Thank you. We have added Bayes factor values for ANOVAs (implemented in JASP[13]) wherever applicable in the revised manuscript. For example:

“While we found a main effect of both lag (detection: F(1,23)=29.8, p<0.001, BF >10^3^ discrimination: F(1,23)=54.1, p<0.001, BF >10^3^) and contrast (detection: F(1,23)=21.02, p<0.001, BF>10^2^, discrimination: F(1,23) = 13.75, p=0.001, BF=1.22), we found no significant interaction effect between lag and contrast (detection: F(1,23)=1.92, p=0.113, BF=0.49, discrimination: F(1,23) = 0.93, p=0.450, BF=0.4).”

“A two-way ANOVA with inter-target lag and T2 contrast as independent factors revealed a main effect of lag on both d’_det_ (F(1,23)=30.3, p<0.001, BF>10^3^) and d’_dis_ (F(1,23)=100.3, p<0.001, BF>10^3^). Yet, we found no significant interaction effect between lag and contrast for d’_det_ (F(1,23)=2.3, p=0.141, BF=0.44).”

Minor points(3.b) Statistics: Many p-values are reported without the respective test statistics (e.g., p. 9, l. 164; p. 12, l. 241-244 and 252-258; p. 13, l. 271, etc.).

Addressed in response to public reviews, #3.7.

(3.c) P. 4, l. 58: It is not entirely clear how the authors define "early or late". For example, while they consider the P2/N2/N2pc complex as "late" (l. 62-64), these ERP components are considered "early" in the debate on "early vs. late" neural correlates of consciousness (for a review, see Förster et al., 2020).

We appreciate the debate. Our naming convention follows these seminal works[3,14–16].

(3.d) P. 5., l. 77: "previous studies have essentially treated the attentional blinks as a unitary, monolithic phenomenon": There are previous studies in which both the presence and identity of T2 were queried (e.g., Eiserbeck et al., 2022; Harris et al., 2013).

Addressed in response to recommendations for authors, #1.a.

(3.e) P. 9, l. 169-177: The detection and discrimination accuracies are analyzed using twoway ANOVAs with the factors lags and contrast. I wonder why the lag effects are additionally analyzed using Wilcoxon signed rank tests using data pooled across the T2 contrasts (p., 9, l. 161-168)? If I understand it correctly, these tests should correspond to the main effects of lag in the ANOVAs. Indeed, both analyses lead to the same conclusions (l. 167 and l. 176).

Our motivation was to first establish the attentional blink effect, with data pooled across contrasts. The subsequent ANOVA allowed delving deeper into contrast and interaction effects. Indeed, the results were consistent across both tests.

(3.f) P. 12, l. 242: I wonder why the T2 contrasts are pooled in the statistical tests (but plotted separately, p. 45, Figure 3C).

Model selection analysis distinct d’_det_ parameter values across contrasts, as reflected in Fig. 3C. As mentioned in response #3.e contrasts effects were analyzed with an ANOVA.

(3.g) P. 13, l. 287: "high and low contrast T2 trials were pooled to estimate reliable ERPs". The amount of trials per condition is not provided.

Addressed in response to public reviews, #3.7.

(3.h) P. 45, Figure 3D/F: In my opinion, plotting the contrasts and lags separately (despite the results of the model selection) would have provided a better idea of the data.

We appreciate the reviewer’s suggestion, but followed the results of model selection for consistency.

(3.i) P. 21, l. 470: "the left index finger to report clockwise orientations and the right index finger to report counter-clockwise orientations": This left/right mapping seems counterintuitive to me, and the authors also used the opposite mapping in Figures 1 and 2. It is not described in the Methods (p. 25) and thus is unclear.

We regret the typo. Revised as follows:

“...the left index finger to report counter-clockwise orientations and the right index finger to report clockwise orientations.”

(3.j) P. 22, l. 514: "Taken together, these results suggest the following, testable schema (SI Figure S5)." Figure S5 seems to be missing.

Amended. This is Fig. 8 in the revised manuscript.

(3.k) P. 25, l. 559: I do not understand why the circular placeholders around the stimuli were included, and they are not mentioned in Figure 2A (p. 43). When I saw the figure and read the inscription, I wondered whether they were actually part of the stimulus presentation or symbolized something else.

The placeholder was described in the earlier Methods section. We have now also mentioned it in caption for Fig. 2A.

“All plaids were encircled by a circular placeholder. The fixation dot and the placeholder were present on the screen throughout the trial.”

This avoided spatial uncertainty with estimating stimulus dimensions during the presentation.

(3.l) P. 32, l. 754: The interval of interest for the P1 from 40 to 140 ms seems unusually early to me. The component usually peaks at 100 ms (e.g., at 96 ms in the cited study by Sergent et al., 2005), which also seems to be the case in the present study (Fig. S3, p. 57). I wonder how they were defined.

For our analyses, we employed the peak value of the P1 ERP component in a window from 40-140 ms. The peak occurred around 100 ms (SI Fig. S3), which aligns with the literature.

Additional minor comments:These comments have been all addressed, and typos corrected, by revising the manuscript at the appropriate places.3.m.1. L. 14: In my opinion, this sentence is difficult to read due to the nested combination of singular and plural forms. Importantly, as the authors also acknowledge (e.g., l. 83), perceptual sensitivity and choice bias could both be compromised, so I would suggest using plural and adding "or both" as a third option for clarity. See also p. 10, l. 204.3.m.2. L. 14: The comma before "As a result" should be replaced by a period.3.m.3. L. 45 "to guide Behavior" should be lowercase.3.m.4. L. 67: "Activity in the parietal, lateral prefrontal cortex and anterior cingulate cortex" could be read as if there was a "parietal, prefrontal cortex", so I would suggest removing the first "cortex".

Revised/amended.

3.m.5. L. 77: "fundamental questions regarding the component mechanisms of the attentional blink remain unanswered": The term "component mechanisms" is a bit unclear to me.

We elaborate on this term in the very next set of paragraphs in the Introduction.

3.m.6. L. 88: "a lower proportion of correct T2 detections can arise from a lower detection d'". "Arise from" sounds a bit off given that d' is a function of hits and false alarms.3.m.7. L. 95: I would suggest citing the updated edition of the classic "Detection Theory: A User's Guide" by Hautus, Macmillan & Creelman (2021).3.m.8. L. 102: "a oriented grating" should be "an".3.m.9. L. 126: "key neural markers - a local neural marker (event-related potentials) potentials" should be rephrased/corrected.3.m.10. L. 129: There are inconsistent tenses (mostly past tense but "we synthesize").3.m.11. L. 138: Perhaps the abbreviations (e.g., dva, cpd) should be introduced here (first mention) rather than in the Methods below.3.m.12. L. 148: "at the end of each trial participants first, indicated": The comma position should be changed.3.m.13. L. 176 "attentional blink-induced both a ...": The hyphen should be removed.3.m.14. L. 396: I think "but neither of them affects" would be better here.3.m.15. L. 383: "Detection deficits were signaled by ERP components such as the occipitoparietal N2p and the parietal P3": In my opinion, "such as" is too vague here.

Revised/amended.

3.m.16. L. 403: "Neurally, improved detection of attended targets is accompanied by (...) higher ERP amplitudes". Given the different mechanisms underlying the ERP, this section would benefit from more details.

Addressed in response to public reviews, #3.4.

3.m.17. L. 924: References 18 and 46 seem to be the same.3.m.18. L. 1181: I think d'det should be d'dis here.3.m.19. L. 1284: "détection" should be "detection".3.m.20. I found some Figure legends a bit confusing. For example, 5E refers to 4E, but 4E refers to 4C.3.m.21. In Figures 4A/B and 6C/D, some conditions are hidden due to the overlap of CIs. Could they be made more transparent?

Revised/amended.

References:

(1) Fook K.Chua. The effect of target contrast on the attentional blink. Perception & Psychophysics 5, 770–788 (2005).

(2) Chmielewski, W. X., Mückschel, M., Dippel, G. & Beste, C. Concurrent information affects response inhibition processes via the modulation of theta oscillations in cognitive control networks. Brain Structure and Function 221, 3949–3961 (2016).

(3) Sergent, C., Baillet, S. & Dehaene, S. Timing of the brain events underlying access to consciousness during the attentional blink. Nature Neuroscience 8, 1391–400 (2005).

(4) Zivony, A. & Lamy, D. What processes are disrupted during the attentional blink? An integrative review of event-related potential research. Psychonomic Bulletin & Review29, 394–414 (2022).

(5) Pernet, C. R., Wilcox, R. & Rousselet, G. A. Robust Correlation Analyses: False Positive and Power Validation Using a New Open Source Matlab Toolbox. Frontiers in Psychology 3, (2013).

(6) Gross, J. et al. Modulation of long-range neural synchrony reflects temporal limitations of visual attention in humans. PNAS 101, 13050–13055 (2004).

(7) Eric Maris and Robert Oostenveld. Nonparametric statistical testing of EEG and MEG data. Journal of Neuroscience Methods 164, 177–190 (2007).

(8) Hommel, B. & Akyürek, E. G. Lag-1 sparing in the attentional blink: Benefits and costs of integrating two events into a single episode. The Quarterly Journal of Experimental Psychology Section A 58, 1415–1433 (2005).

(9) Livesey, E. J. & Harris, I. M. Target sparing effects in the attentional blink depend on type of stimulus. Attention Perception & Psychophysics 73, 2104–2123 (2011).

(10) Dellert, T. et al. Neural correlates of consciousness in an attentional blink paradigm with uncertain target relevance. Neuroimage 264, 119679 (2022).

(11) Nobre, A., Correa, A. & Coull, J. The hazards of time. Current Opinion in Neurobiology 17, 465– 470 (2007).

(12) Bang, J. W. & Rahnev, D. Stimulus expectation alters decision criterion but not sensory signal in perceptual decision making. Scientific Reports 7, 17072 (2017).

(13) JASP Team. JASP (version 0.19.0.) [Computer Software]. Preprint at (2022).

(14) Luck, S. J. Electrophysiological Correlates of the Focusing of Attention within Complex Visual Scenes: N2pc and Related ERP Components. (Oxford University Press, 2011). doi:10.1093/oxfordhb/9780195374148.013.0161.

(15) Brydges, C. R., Fox, A. M., Reid, C. L. & Anderson, M. Predictive validity of the N2 and P3 ERP components to executive functioning in children: a latent-variable analysis. Frontiers in Human Neuroscience 8, (2014).

(16) Michalewski, H. J., Prasher, D. K. & Starr, A. Latency variability and temporal interrelationships of the auditory event-related potentials (N1, P2, N2, and P3) in normal subjects. Electroencephalography and Clinical Neurophysiology/Evoked Potentials Section 65, 59–71 (1986).